# Generalization in Federated Learning:
# A Conditional Mutual Information Framework

**Ziqiao Wang** [1]    **Cheng Long** [2]    **Yongyi Mao** [3]

## Abstract

Federated learning (FL) is a widely adopted privacy-preserving distributed learning framework, yet its generalization performance remains less explored compared to centralized learning. In FL, the generalization error consists of two components: the out-of-sample gap, which measures the gap between the empirical and true risk for participating clients, and the participation gap, which quantifies the risk difference between participating and non-participating clients. In this work, we apply an information-theoretic analysis via the conditional mutual information (CMI) framework to study FL's two-level generalization. Beyond the traditional supersample-based CMI framework, we introduce a superclient construction to accommodate the two-level generalization setting in FL. We derive multiple CMI-based bounds, including hypothesis-based CMI bounds, illustrating how privacy constraints in FL can imply generalization guarantees. Furthermore, we propose fast-rate evaluated CMI bounds that recover the best-known convergence rate for two-level FL generalization in the small empirical risk regime. For specific FL model aggregation strategies and structured loss functions, we refine our bounds to achieve improved convergence rates with respect to the number of participating clients. Empirical evaluations confirm that our evaluated CMI bounds are non-vacuous and accurately capture the generalization behavior of FL algorithms.

## 1. Introduction

Federated learning (FL) has emerged as the most widely adopted distributed learning framework, enabling multiple (potentially spatially distributed) clients to collaboratively train a global model (McMahan et al., 2017; Yang et al., 2019; Li et al., 2020; Kairouz et al., 2021). Unlike centralized learning, FL utilizes the computational resources of all participating clients while avoiding the need to aggregate all data in a single location before training. This not only reduces the resource overhead associated with data collection but also enhances privacy protection.

In an FL framework, a central server distributes a global model to all participating clients for local training. In contrast to centralized learning—where data typically follow a single distribution—local data on different clients often arise from heterogeneous distributions. The server then aggregates the updated local models to form a new global model, which is redistributed for further training; this iterative process continues until convergence. While the convergence properties of FL have been extensively studied, such as in (Karimireddy et al., 2020; Li et al., 2020; Mitra et al., 2021; Yun et al., 2021; Wang & Ji, 2022), the study of its generalization performance has only recently gained attention (Mohri et al., 2019; Chen et al., 2021; Yagli et al., 2020; Barnes et al., 2022; Hu et al., 2023; Huang et al., 2023; Sefidgaran et al., 2022; 2024; Sun et al., 2024; Gholami & Seferoglu, 2024) and remains relatively underexplored. Moreover, most FL generalization studies focus only on the generalization error of participating clients, overlooking the model's ability to generalize to unseen clients.

Recently, Yuan et al. (2022) introduces a two-level generalization framework, where client distributions are drawn from a meta-distribution. Under this framework, the generalization error is decomposed into two components: the *out-of-sample gap*, measuring the gap between empirical and true risk for participating clients, and the *participation gap*, capturing the difference in true risks between participating and non-participating clients. This broader definition extends the commonly used out-of-sample gap and is especially relevant in cross-device FL scenarios, where clients are sampled from a large and dynamic population.

[1]School of Computer Science and Technology, Tongji University, Shanghai, China [2]Department of Applied Physics and Applied Mathematics, Columbia University, New York, USA [3]School of Electrical Engineering and Computer Science, University of Ottawa, Ottawa, Canada. Correspondence to: Ziqiao Wang <ziqiaowang@tongji.edu.cn>.

*Proceedings of the 42nd International Conference on Machine Learning*, Vancouver, Canada. PMLR 267, 2025. Copyright 2025 by the author(s).

Among many analytic tools, information-theoretic generalization analysis, pioneered by (Russo & Zou, 2016; 2019; Xu & Raginsky, 2017), has the advantage of accounting for both the data distribution and the learning algorithm, often giving tighter generalization bounds compared to distribution-independent or algorithm-independent methods. The original mutual information (MI)-based generalization bounds in (Xu & Raginsky, 2017; Bu et al., 2020) have been applied to FL in (Yagli et al., 2020; Barnes et al., 2022; Zhang et al., 2024). However, it is well-known that these MI-based bounds suffer from unboundedness although the true generalization error can be small (Bassily et al., 2018). To address this issue, Steinke & Zakynthinou (2020) proposes the conditional mutual information (CMI) framework, which introduces a "supersample" construction. Here, an auxiliary "ghost sample" is drawn alongside the original training sample, and a sequence of Bernoulli random variables determines which data points are used for training. The CMI between these Bernoulli variables and the hypothesis (e.g., model parameters), conditioned on the supersample, serves as a generalization measure. This construction ensures bounded CMI terms due to bounded entropy of Bernoulli variables, inherently leading to tighter generalization bounds compared to the standard MI-based methods, as shown in Haghifam et al. (2020). The CMI framework has seen multiple refinements, including fast-rate and numerically tight bounds (Harutyunyan et al., 2021; Hellström & Durisi, 2022a; Wang & Mao, 2023a;c; 2024b; Dong et al., 2024). Despite the success of these CMI bounds, their effectiveness in capturing FL generalization is still not established.

In this work, we provide the first CMI generalization framework for FL in the two-level generalization setting. Inspired by the supersample-based CMI framework, we introduce an additional "superclient" construction, where clients are grouped into two sets, and Bernoulli random variables determine their participation. This symmetric structure enables a CMI-based characterization of the participation gap, while the out-of-sample gap is analyzed using standard supersample-based CMI techniques as in Steinke & Zakynthinou (2020). Our main contributions are as follows:

- Based on our superclient and supersamples construction, we derive the first CMI-based generalization bound (cf. Theorem 3.1) for FL, consisting of two terms: (i) the CMI between the hypothesis and the Bernoulli variable governing client participation, and (ii) the CMI between the hypothesis and the Bernoulli variable governing training data membership. These terms reveal that FL models generalize well to unseen clients when they leak minimal information about training data membership and client participation. Furthermore, we show that differential privacy constraints at both local and global levels naturally imply gen-

eralization (cf. Lemma 3.2). In the special case of identical client data distributions, the bound reduces to a single CMI term (cf. Corollary 3.2), aligning with intuition. We also extend our framework to high-probability bounds (cf. Theorem 3.2) and excess risk analysis (cf. Theorem 3.3).

- To improve the bound's convergence rate, we derive evaluated CMI (e-CMI) bounds (cf. Theorem 4.1), using techniques from Wang & Mao (2023c). These bounds not only recover the best-known FL convergence rate in the low empirical risk regime, as shown in Hu et al. (2023), but also provide significant practical advantages. Unlike hypothesis-based CMI bounds that involve high-dimensional random variables, e-CMI bounds only require computing CMI between one-dimensional random variables, making them much easier to estimate in practice.

- We extend our analysis to scenarios where model aggregation strategies (e.g., model averaging) and structured loss functions play a role. Specifically, following Barnes et al. (2022), we derive a CMI bound under Bregman loss and show that both the participation gap and out-of-sample gap exhibit fast-rate convergence with respect to the number of participating clients (cf. Theorem 5.1). Further, inspired by Gholami & Seferoglu (2024), we establish a sharper CMI bound under strongly convex and smooth loss functions (cf. Theorem 5.2), demonstrating even faster decay rates for out-of-sample gap.

- To verify our results, we conduct FL experiments using FedAvg (McMahan et al., 2017) on two datasets. We show that our e-CMI bounds are numerically nonvacuous and effectively capture the generalization behavior of FL.

## 2. Preliminaries

**Notations** Throughout this paper, unless otherwise stated, we use capital letters (e.g., $X$) to denote random variables and the corresponding lowercase letters (e.g., $x$) to denote realized values. Let $P_X$ be the distribution of $X$, and $P_{X|Y}$ the conditional distribution of $X$ given $Y$. When conditioning on a specific realization, we write $P_{X|Y=y}$ or simply $P_{X|y}$. We also use $\mathbb{E}_X$ and $\mathbb{E}_{P_X}$ interchangeably to denote the expectation over $X \sim P_X$, whenever the underlying distribution is clear. Similarly, $\mathbb{E}_{X|Y=y}$ (or $\mathbb{E}_{X|y}$) denotes the expectation with respect to $X \sim P_{X|Y=y}$. Let $\mathrm{D_{KL}}(P||Q)$ be the Kullback–Leibler (KL) divergence between $P$ and $Q$. Let $I(X;Y) \triangleq \mathrm{D_{KL}}(P_{X,Y}||P_X P_Y)$ be the mutual information (MI) between $X$ and $Y$, and $I(X;Y|Z)$ the conditional mutual information (CMI) between $X$ and $Y$ given $Z$. Following Ne-

grea et al. (2019), we define the disintegrated mutual information for $I^z(X;Y) \triangleq D_{KL}(P_{X,Y|Z=z}||P_{X|Z=z}P_{Y|Z=z})$, and note that $I(X;Y|Z) = \mathbb{E}_Z\left[I^Z(X;Y)\right]$.

**Federated Learning Problem Setup**   We consider a federated learning (FL) setup with $K$ *participating* clients. Let $\mathcal{Z}$ be the instance space, $\mathcal{W}$ the hypothesis space, and $\mathcal{C}$ a (possibly infinite) set of all *potential* clients. Each client $c \in \mathcal{C}$ has a local data distribution $\mu_c$ on $\mathcal{Z}$. Following Yuan et al. (2022), we assume that there is a distribution $\mathcal{D}$ on $\mathcal{C}$, referred to as "meta-distribution" in Yuan et al. (2022), and that $K$ participating clients $c_1, c_2, \ldots, c_K$ are drawn independently from $\mathcal{D}$. We will write client $c_i$ ($i \in [K]$) simply as $i$ for simplicity, e.g., write the distribution $\mu_{c_i}$ of client $i$ as $\mu_i$. Let $S_i = \{Z_{i,j}\}_{j=1}^n$ with $Z_{i,j} \overset{i.i.d}{\sim} \mu_i$ denote the training dataset for client $i$, where for simplicity we have assumed that the training sets have the same size across all clients. In FL, each client $i$ applies a local algorithm $\mathcal{A}_i : \mathcal{Z}^n \to \mathcal{W}$, described by the conditional distribution $P_{W_i|S_i}$, to produce a local model $W_i = \mathcal{A}_i(S_i)$. Note that $\mathcal{A}_i$ may differ across clients. These local models $\{W_i\}_{i=1}^K$ are then transmitted to a central server, which applies an aggregation algorithm to produce a global model $W$. Hence, an FL algorithm $\mathcal{A}$ is composed of all the local algorithms together with the model aggregation procedure. Denote $S = \cup_{i \in [K]} S_i$. The overall FL algorithm $\mathcal{A}$ is then conceptually characterized by a conditional distribution $P_{W|S}$, which takes the collective training sample $S$ as input and outputs a hypothesis $W \in \mathcal{W}$. Formally, $\mathcal{A} : \mathcal{Z}^{nK} \to \mathcal{W}$. We measure the quality of $W$ using a loss function $\ell : \mathcal{W} \times \mathcal{Z} \to \mathbb{R}_0^+$.

**Generalization Error**   We adopt the two-level generalization framework introduced in (Yuan et al., 2022; Hu et al., 2023), where the ultimate goal in FL is set to minimizing the true risk for unseen or non-participating clients. Concretely, for any hypothesis $w \in \mathcal{W}$, the *global true risk* (or population risk) is defined as

$$L_{\mathcal{D}}(w) \triangleq \mathbb{E}_{\mu \sim \mathcal{D}} \mathbb{E}_{Z \sim \mu} [\ell(w, Z)].$$

For the $i$-th participating client, we define its true risk and empirical risk as $L_{\mu_i}(w) \triangleq \mathbb{E}_{Z \sim \mu_i} [\ell(w, Z)]$ and $L_{S_i}(w) \triangleq \frac{1}{n} \sum_{j=1}^n \ell(w, Z_{i,j})$, respectively. For the sake of simplicity, we treat each client equally, then we define the *average client true risk* for the participating clients as

$$L_{\mu_{[K]}}(w) \triangleq \frac{1}{K} \sum_{i=1}^K L_{\mu_i}(w) = \frac{1}{K} \sum_{i=1}^K \mathbb{E}_{Z'_i \sim \mu_i} [\ell(w, Z'_i)],$$

where $Z'_i \sim \mu_i$ is an independent testing instance from the $i$-th participating client's distribution $\mu_i$. Likewise, the *average client empirical risk* is given by

$$L_S(w) \triangleq \frac{1}{K} \sum_{i=1}^K L_{S_i}(w) = \frac{1}{Kn} \sum_{i=1}^K \sum_{j=1}^n \ell(w, Z_{i,j}),$$

which serves as a practical proxy for the average client true risk of $w$ because data distributions are unknown in real scenarios. Finally, for an FL algorithm $\mathcal{A}$, we define its expected generalization error as

$$\mathcal{E}_{\mathcal{D}}(\mathcal{A}) \triangleq \mathbb{E}_W [L_{\mathcal{D}}(W)] - \mathbb{E}_{W,S} [L_S(W)].$$

**CMI Framework for FL**   We now adapt the *supersample* construction of Steinke & Zakynthinou (2020) to an FL setting by introducing a *superclient* and corresponding supersamples for each client. Let $\tilde{\mu}$ be a $K \times 2$ matrix, namely the superclient, whose entries are drawn independently from the meta-distribution $\mathcal{D}$. We index the columns of $\tilde{\mu}$ by $\{0,1\}$ and denote the $i$-th row by $\tilde{\mu}_i = (\tilde{\mu}_{i,0}, \tilde{\mu}_{i,1})$. Next, we construct a supersample for each client distribution in $\tilde{\mu}$. For example, the supersample $\widetilde{Z}^{i,0} \in \mathcal{Z}^{n \times 2}$ for $\tilde{\mu}_{i,0}$ has its entries drawn i.i.d. from $\tilde{\mu}_{i,0}$. We index the columns of $\widetilde{Z}^{i,0}$ by $\{0,1\}$, and write $\widetilde{Z}_j^{i,0} = (\widetilde{Z}_{j,0}^{i,0}, \widetilde{Z}_{j,1}^{i,0})$ for the $j$-th row. The same construction applies for all the other client distributions in $\tilde{\mu}$. Thus, we have one superclient matrix $\tilde{\mu}$ and $2K$ supersample matrices in total. To determine which client distributions participate in the training, we introduce a random variable $V = \{V_i\}_{i=1}^K$, where each $V_i$ is drawn i.i.d. from $\mathrm{Unif}(\{0,1\})$ and is independent of $\tilde{\mu}$. If $V_i = 0$, then the client distribution $\tilde{\mu}_{i,0}$ is included in the participating client set, and $\tilde{\mu}_{i,1}$ is non-participating; if $V_i = 1$, the opposite holds. For each client distribution, we further introduce $U^{i,b} = \{U_j^{i,b}\}_{j=1}^n$ (with $b \in \{0,1\}$), where each $U_j^{i,b} \overset{i.i.d.}{\sim} \mathrm{Unif}(\{0,1\})$ is independent of $\widetilde{Z}^{i,b}$. These variables specify which column in the supersample is used for training versus testing. For instance, when $U_j^{i,0} = 0$, $\widetilde{Z}_{j,0}^{i,0}$ is part of the training set, and $\widetilde{Z}_{j,1}^{i,0}$ is used for local testing; if $U_j^{i,0} = 1$, these roles are reversed. Let $\overline{V}_i = 1 - V_i$. The set of participating client distributions is then $\tilde{\mu}_V = \{\tilde{\mu}_{i,V_i}\}_{i=1}^K$, and the set of corresponding supersamples is denoted by $\widetilde{Z}^V = \{\widetilde{Z}^{i,V_i}\}_{i=1}^K$, while the non-participating client distributions are $\tilde{\mu}_{\overline{V}} = \{\tilde{\mu}_{i,\overline{V}_i}\}_{i=1}^K$ with supersamples $\widetilde{Z}^{\overline{V}}$. Similarly, for each participating distribution $\tilde{\mu}_{i,b}$, define $\overline{U}_j^{i,b} = 1 - U_j^{i,b}$. Its training sample (i.e. $S_i$) is $\widetilde{Z}_{U^{i,b}}^{i,b} = \{\widetilde{Z}_{j,U_j^{i,b}}^{i,b}\}_{j=1}^n$, and the corresponding test sample is $\widetilde{Z}_{\overline{U}^{i,b}}^{i,b} = \{\widetilde{Z}_{j,\overline{U}_j^{i,b}}^{i,b}\}_{j=1}^n$.

Notably, the supersamples for the non-participating clients, although well defined, are actually irrelevant, as none of those data are used in training. To improve readability, Appendix A provides a visualization of our superclient and supersample construction (cf. Figure 2), along with a summary of notations (cf. Table 1).

## 3. Hypothesis-based CMI Bound

As previously discussed, generalization in FL involves two levels: (i) generalizing from participating clients to unseen clients, and (ii) generalizing from the training data to unseen data of the participating clients. Following (Yuan et al., 2022; Hu et al., 2023), we separate these two levels of generalization through the following decomposition:

$$\mathcal{E}_{\mathcal{D}}(\mathcal{A}) = \underbrace{\mathbb{E}_W\left[L_{\mathcal{D}}(W)\right] - \mathbb{E}_{W,\mu_{[K]}}\left[L_{\mu_{[K]}}(W)\right]}_{\mathcal{E}_{PG}(\mathcal{A}):\text{Participation Gap}}$$
$$+ \underbrace{\mathbb{E}_{W,\mu_{[K]}}\left[L_{\mu_{[K]}}(W)\right] - \mathbb{E}_{W,S}\left[L_S(W)\right]}_{\mathcal{E}_{OG}(\mathcal{A}):\text{Out-of-Sample Gap}}. \quad (1)$$

The *participation gap*, denoted as $\mathcal{E}_{PG}(\mathcal{A})$, quantifies the difference in test performance between non-participating and participating clients. The *out-of-sample gap* (also called participation error in (Hu et al., 2023)), denoted as $\mathcal{E}_{OG}(\mathcal{A})$, represents the average of the local generalization gaps over the $K$ participating clients. Note that most existing FL generalization studies focus primarily on $\mathcal{E}_{OG}(\mathcal{A})$ (Yagli et al., 2020; Barnes et al., 2022; Chor et al., 2023; Sefidgaran et al., 2022; 2024; Sun et al., 2024).

Under our construction of the superclient and supersamples in FL, the following lemma will be handy in our analysis.

**Lemma 3.1.** *The participation gap $\mathcal{E}_{PG}(\mathcal{A})$ can be rewritten as*

$$\frac{1}{K}\sum_{i=1}^{K}\mathbb{E}\left[(-1)^{V_i}\left(\ell(W,\widetilde{Z}_{1,\overline{U}_1^{i,1}}^{i,1}) - \ell(W,\widetilde{Z}_{1,\overline{U}_1^{i,0}}^{i,0})\right)\right],$$

*where the expectation is taken over $P_{\widetilde{Z}^i,U^i,W,V_i}$, $\widetilde{Z}^i = (\widetilde{Z}^{i,0},\widetilde{Z}^{i,1})$ and $U^i = (U^{i,0},U^{i,1})$.*

*The out-of-sample gap $\mathcal{E}_{OG}(\mathcal{A})$ can be rewritten as*

$$\frac{1}{Kn}\sum_{i=1}^{K}\sum_{j=1}^{n}\mathbb{E}\left[(-1)^{U_j^{i,V_i}}\left(\ell(W,\widetilde{Z}_{j,1}^{i,V_i}) - \ell(W,\widetilde{Z}_{j,0}^{i,V_i})\right)\right],$$

*where the expectation is taken over $P_{\widetilde{Z}^i,V_i,U_j^{i,V_i},W}$.*

**Remark 3.1.** *Notably, by the symmetric property of superclient and supersamples, Lemma 3.1 indicates that $\frac{1}{K}\sum_{i=1}^{K}(-1)^{V_i}\left(\ell(W,\widetilde{Z}_{1,\overline{U}_1^{i,1}}^{i,1}) - \ell(W,\widetilde{Z}_{1,\overline{U}_1^{i,0}}^{i,0})\right)$ and $\frac{1}{Kn}\sum_{i=1}^{K}\sum_{j=1}^{n}(-1)^{U_j^{i,V_i}}\left(\ell(W,\widetilde{Z}_{j,1}^{i,V_i}) - \ell(W,\widetilde{Z}_{j,0}^{i,V_i})\right)$ are unbiased estimators for $\mathcal{E}_{PG}(\mathcal{A})$ and $\mathcal{E}_{OG}(\mathcal{A})$, respectively. We remark that the subscript index "1" in $\widetilde{Z}_{1,\overline{U}_1^{i,1}}^{i,1}$ and $\widetilde{Z}_{1,\overline{U}_1^{i,0}}^{i,0}$ can be replaced with any other $j \in [n]$, as the participation gap does not depend on the order of elements in the testing datasets. Furthermore, if the algorithm is symmetric—meaning $W$ does not depend on the order of elements in the local training sets, a common assumption used in stability-based generalization analysis (Bousquet & Elisseeff, 2002)—then the averaging over $n$ data points in $\mathcal{E}_{OG}(\mathcal{A})$ can also be omitted.*

### 3.1. First CMI Bound

The bounding steps for $\mathcal{E}_{\mathcal{D}}(\mathcal{A})$ closely follow those in standard centralized analysis, once Lemma 3.1 has been established.

We are now in a position to present the first CMI bound for FL.

**Theorem 3.1.** *Assume $\ell(\cdot,\cdot) \in [0,1]$, the following bound holds*

$$|\mathcal{E}_{\mathcal{D}}(\mathcal{A})| \le \frac{1}{K}\sum_{i=1}^{K}\mathbb{E}_{\widetilde{Z}^i,U^i}\sqrt{2I^{\widetilde{Z}^i,U^i}(W;V_i)}$$
$$+ \frac{1}{Kn}\sum_{i=1}^{K}\sum_{j=1}^{n}\mathbb{E}_{\widetilde{Z}^i,V_i}\sqrt{2I^{\widetilde{Z}^i,V_i}(W;U_j^{i,V_i})}.$$

The bound consists of two terms, each providing an upper bound for $\mathcal{E}_{PG}(\mathcal{A})$ and $\mathcal{E}_{OG}(\mathcal{A})$, respectively. Notably, both CMI terms in the bound preserve the properties of the standard CMI from Steinke & Zakynthinou (2020). For example, $I^{\widetilde{z}^i,u^i}(W;V_i) \le H(V^i) = \log 2$ and $I^{\widetilde{z}^i,v_i}(W;U_j^{i,V_i}) \le H(U_j^{i,V_i}) = \log 2$, where $H(\cdot)$ denotes the Shannon entropy (Thomas & Joy, 2006). Consequently, unlike previous MI-based FL generalization bounds (Yagli et al., 2020; Barnes et al., 2022; Zhang et al., 2024) that can grow unbounded (see Bassily et al. (2018)), our CMI-based bound in Theorem 3.1 is strictly bounded.

Furthermore, if both CMI terms are zero, then the algorithm's output is independent of its input (i.e., independent of the chosen participating clients and their local training data). In contrast, if they achieve their upper bounds, the algorithm reveals all membership information about the clients and their local training sets. Specifically, the first CMI term quantifies how well one can infer the membership of the "participating client set" from the output hypothesis when $(\tilde{z}^i, u^i)$ are known, whereas the second term quantifies how well one can infer the membership of the "local training set" when $(\tilde{z}^i, v_i)$ are given. Clearly, if the FL algorithm $\mathcal{A}$ enforces privacy constraints that make these inferences (i.e. determining $V_i$ and $U_j^{i,V_i}$) difficult, then both CMI terms remain small, resulting in a small generalization error; in other words, privacy implies generalization in FL. Inspired by (Cuff & Yu, 2016; Barnes et al., 2022), when FL algorithms are differentially private (Dwork et al., 2006a;b), this connection is formally established in the following lemma.

**Lemma 3.2.** *If each local algorithm $\mathcal{A}_i$ is $\epsilon_i$-differentially private, and the overall FL algorithm $\mathcal{A}$ is $\epsilon'$-differentially*

*private, then*

$$\frac{1}{K}\sum_{i=1}^{K}\mathbb{E}_{\widetilde{Z}^i,U^i}\sqrt{I^{\widetilde{Z}^i,U^i}(W;V_i)} \leq \sqrt{\frac{\min\{\epsilon',(e^{\epsilon'}-1)\epsilon'\}}{K}},$$

$$\frac{1}{n}\sum_{j=1}^{n}\mathbb{E}_{\widetilde{Z}^i,V_i}\sqrt{I^{\widetilde{Z}^i,V_i}(W;U_j^{i,V_i})} \leq \sqrt{\frac{\min\{\epsilon_i,(e^{\epsilon_i}-1)\epsilon_i\}}{n}}.$$

**Remark 3.2.** *Combining Lemma 3.2 with Theorem 3.1 implies the bound* $|\mathcal{E}_{\mathcal{D}}(\mathcal{A})| \leq \mathcal{O}\left(\sqrt{\frac{\epsilon'}{K}} + \frac{1}{K}\sum_{i=1}^{K}\sqrt{\frac{\epsilon_i}{n}}\right)$, *showing that $\mathcal{A}$ admits a valid generalization guarantee under differential privacy. We remark that it is possible to obtain $\epsilon'$ using all $\epsilon_i$ for certain FL algorithm, as studied in Kairouz et al. (2015). If we let $\epsilon_i = \epsilon$ for all $i$, then we obtain $|\mathcal{E}_{\mathcal{D}}(\mathcal{A})| \leq \mathcal{O}\left(\sqrt{\frac{\epsilon'}{K}} + \sqrt{\frac{\epsilon}{n}}\right)$. Barnes et al. (2022) also provides a privacy-based result bounding $\mathcal{E}_{OG}(\mathcal{A})$, showing that when certain model aggregation strategies and loss functions are used, $\mathcal{E}_{OG}(\mathcal{A}) \leq \mathcal{O}\left(\frac{1}{K}\sqrt{\frac{\epsilon}{n}}\right)$. We will discuss their setting in Section 5.*

Let $\widetilde{Z} = \{\widetilde{Z}^i\}_{i=1}^{K}$ be the collection of all supersamples and $U = \{U^i\}_{i=1}^{K}$ be the collection of all Bernoulli variables for determining sample usage. The following CMI bound follows from Jensen's inequality and the chain rule of MI.

**Corollary 3.1.** *Assume $\ell(\cdot,\cdot) \in [0,1]$, then the following bound holds*

$$|\mathcal{E}_{\mathcal{D}}(\mathcal{A})| \leq \sqrt{\frac{2I(W;V|\widetilde{Z},U)}{K}} + \sqrt{\frac{2I(W;U|\widetilde{Z},V)}{Kn}}.$$

Hence, we achieve a rate of order[1] $\mathcal{O}\left(\frac{1}{\sqrt{K}} + \frac{1}{\sqrt{Kn}}\right)$. Recently, Zhang et al. (2024, Theorem 5.1) presents a MI-based FL bound of the form $\mathcal{O}\left(\sqrt{\frac{I(W;\mu_{[K]})}{K}} + \sqrt{\frac{I(W;S)}{Kn}}\right)$, which is the input-output mutual information (IOMI) (Xu & Raginsky, 2017) version of our Corollary 3.1. Since CMI is always no larger than the corresponding IOMI counterpart (Haghifam et al., 2020, Theorem 2.1), our bound is always tighter than Zhang et al. (2024, Theorem 5.1). Additionally, as a by-product, we also provide a novel IOMI bound for FL in Theorem C.2 in Appendix.

Before moving on, let us consider the i.i.d. (homogeneous) FL setting. Intuitively, if all clients share the same data distribution, then the first CMI term—which bounds $\mathcal{E}_{PG}(\mathcal{A})$—should vanish. In particular, when each $\mu_i$ is identical, $V_i$ no longer influences the algorithm's output distribution once $\widetilde{Z}^i$ and $U^i$ are given. The corollary below formalizes this intuition.

---

[1] In this paper, due to the absence of explicit decay rates for the CMI or MI terms, we often follow previous works by simply treating these terms as $\mathcal{O}(1)$ when stating order-wise behavior. However, it should be noted that the actual decay of the bound is likely to be faster than the rate presented.

**Corollary 3.2.** *If $|\mathcal{C}| = 1$, then we have*

$$|\mathcal{E}_{\mathcal{D}}(\mathcal{A})| \leq \sqrt{\frac{2I(W;U|\widetilde{Z})}{Kn}}.$$

Corollary 3.2 recovers the standard CMI bound of Steinke & Zakynthinou (2020) when the total dataset size is $Kn$. It also highlights that the first term in Corollary 3.1 quantifies the effect of heterogeneity on FL generalization.

### 3.2. High Probability CMI Bounds

Our previous bounds are all provided on the expected generalization error. However, classical learning theory often focuses on high-probability (PAC-style) bounds (Shalev-Shwartz & Ben-David, 2014). In what follows, we establish a high-probability CMI-based generalization bound for FL. Specifically, the theorem below is the high-probability analog of Corollary 3.1, featuring a square-root dependence.

**Theorem 3.2.** *Assume $\ell(\cdot,\cdot) \in [0,1]$, and let $P_{W|\widetilde{Z},U,V}$ be the the conditional distribution of $W$ given $(\widetilde{Z},U,V)$. Let $P_{W|\widetilde{Z},U} = \mathbb{E}_V\left[P_{W|\widetilde{Z},U,V}\right]$ and let $P_{W|\widetilde{Z},V} = \mathbb{E}_U\left[P_{W|\widetilde{Z},V,U}\right]$. Then, with probability at least $1-\delta$ under the draw of $(\widetilde{Z},U,V)$, the generalization error $\left|\mathbb{E}_{W|\widetilde{Z},U,V}\left[L_{\mathcal{D}}(W)-L_S(W))\right]\right|$ is upper bounded by*

$$\sqrt{\frac{2\mathrm{D}_{\mathrm{KL}}\left(P_{W|\widetilde{Z},U,V}||P_{W|\widetilde{Z},U}\right) + 2\log\frac{\sqrt{K}}{\delta}}{K-1}}$$
$$+\sqrt{\frac{2\mathrm{D}_{\mathrm{KL}}\left(P_{W|\widetilde{Z},U,V}||P_{W|\widetilde{Z},V}\right) + 2\log\frac{\sqrt{Kn}}{\delta}}{Kn-1}}.$$

Notice that the second term in the bound has the rate $\mathcal{O}(\frac{1}{\sqrt{Kn}})$ matching the PAC-Bayesian bound of Sefidgaran et al. (2024, Theorem 3.1) for a bounded loss.

### 3.3. Excess Risk Bound

In addition to standard generalization error, we now study the excess risk for FL. Specifically, let $w^* = \arg\min_{w \in \mathcal{W}}\mathbb{E}_{\mu\sim\mathcal{D}}\mathbb{E}_{Z\sim\mu}\left[\ell(w,Z)\right]$ be a global risk minimizer under the meta-distribution $\mathcal{D}$. Note that $w^*$ may not be unique. The expected excess risk is then

$$\mathcal{E}_{ER}(\mathcal{A}) \triangleq \mathbb{E}_W\left[L_{\mathcal{D}}(W)\right] - \mathbb{E}_{\mu\sim\mathcal{D}}\mathbb{E}_{Z\sim\mu}\left[\ell(w^*,Z)\right].$$

Moreover, we focus on the ERM setting, i.e., the FL algorithm $\mathcal{A}$ outputs a hypothesis $W$ that minimizes the empirical risk $L_S(W)$ for the participating clients. The expected generalization bound and high-probability generalization bound for ERM FL algorithm's excess risk are given next.

**Theorem 3.3.** *Assume $\ell(\cdot, \cdot) \in [0,1]$, then for any ERM FL learning algorithm $\mathcal{A}$, the following bound holds*

$$\mathcal{E}_{ER}(\mathcal{A}) \leq \sqrt{\frac{2I(W;V|\widetilde{Z},U)}{K}} + \sqrt{\frac{2I(W;U|\widetilde{Z},V)}{Kn}}.$$

*Furthermore, with probability at least $1 - \delta$ under the draw of $(\widetilde{Z}, U, V)$,*

$$\left| \mathbb{E}_{W|\widetilde{Z},U,V}\left[L_{\mathcal{D}}(W) - \mathbb{E}_Z\left[\ell(w^*, Z)\right]\right] \right| \leq \Xi + \sqrt{\frac{\log \frac{2}{\delta}}{2Kn}},$$

*where $\Xi$ is the upper bound given in Theorem 3.2.*

Thus, in the absence of additional assumptions, the excess risk for the ERM algorithm exhibits an $\mathcal{O}(\frac{1}{\sqrt{K}} + \frac{1}{\sqrt{Kn}})$ convergence rate.

## 4. Fast-rate Evaluated CMI Bounds

The above bounds show a rate of $\mathcal{O}(\frac{1}{\sqrt{K}} + \frac{1}{\sqrt{Kn}})$, we now improve this rate. Thanks to the symmetric property of superclient and supersamples, we are able to adopt the single-loss technique from Wang & Mao (2023c). This gives us a variant of the evaluated CMI (e-CMI) bound (Steinke & Zakynthinou, 2020; Hellström & Durisi, 2022a).

**Theorem 4.1.** *Assume $\ell(\cdot, \cdot) \in [0,1]$. Denote the random variables $\bar{L}_i^+ = \ell(W, \widetilde{Z}_{1,\overline{U}_1^{i,0}}^{i,0})$ and $L_j^{i+} = \ell(W, \widetilde{Z}_{j,0}^{i,V_i})$, then there exist constants $C_1, C_2, C_3, C_4 \in \{C_1, C_2 > 1, C_3, C_4 > 0 | e^{-2C_3C_1} + e^{2C_3} \leq 2, e^{-2C_4C_2} + e^{2C_4} \leq 2\}$ such that the following bound holds,*

$$\mathbb{E}_W\left[L_{\mathcal{D}}(W)\right] \leq C_1 C_2 \mathbb{E}_{W,S}\left[L_S(W)\right] + \sum_{i=1}^K \frac{I(\bar{L}_i^+; V_i)}{C_3 K}$$
$$+ \sum_{i=1}^K \sum_{j=1}^n \frac{C_1 I(L_j^{i+}; U_j^{i,V_i}|V_i)}{C_4 Kn}.$$

Notably, the square-root functions are removed from the CMI terms, at the cost of a multiplicative factor $C_1 C_2$ to the average client empirical risk $\mathbb{E}_{W,S}\left[L_S(W)\right]$. Another advantage of the e-CMI bound is that it involves only one-dimensional random variables, unlike the hypothesis-based CMI bound, where $W$ is a high-dimensional random variable. This makes e-CMI significantly easier to estimate in practice. Moreover, by the data-processing inequality, a fast-rate hypothesis-based CMI bound is established below.

**Corollary 4.1.** *Assume $\ell(\cdot, \cdot) \in [0,1]$, then there exist constants $C_1, C_2, C_3, C_4$ satisfying the conditions in Theo-*

*rem 4.1 such that the following bound holds,*

$$\mathbb{E}_W\left[L_{\mathcal{D}}(W)\right] \leq C_1 C_2 \mathbb{E}_{W,S}\left[L_S(W)\right] + \frac{I(W;V|\widetilde{Z},U)}{C_3 K}$$
$$+ \frac{C_1 I(W;U|\widetilde{Z},V)}{C_4 Kn}.$$

When the average empirical risk $\mathbb{E}_{W,S}\left[L_S(W)\right]$ is sufficiently small, this bound suggests a fast rate of order $\mathcal{O}\left(\frac{1}{K} + \frac{1}{Kn}\right)$. Unlike (Barnes et al., 2022; Gholami & Seferoglu, 2024), our fast-rate result does not require any specific loss function structure beyond boundedness. Similar fast-rate behavior of generalization error under additional conditions (e.g., Bernstein condition, uniform entropy on $\mathcal{W}$) is also observed by Hu et al. (2023) for Lipschitz losses. In fact, we expect our fast-rate bounds could be extended (e.g., via the variance-based and sharpness-based analyses in (Wang & Mao, 2023c; Dong et al., 2024) or binary-KL bound in (Hellström & Durisi, 2022a;b; Hellström & Guedj, 2024)), relaxing the need for strictly small empirical risk.

## 5. CMI Bounds for Model Aggregation in FL

In the previous analysis, the learning algorithm $\mathcal{A}$ is treated as a black-box, where only the inputs and outputs of the algorithm are considered. In this section, we extend the analysis to account for interactions and communications between clients by incorporating model aggregation in FL. Here, we follow the setting in Barnes et al. (2022), where the aggregation is simply the average of all local models, namely $W = \frac{1}{K}\sum_{i=1}^K W_i$, which corresponds to the FedAvg algorithm (McMahan et al., 2017). Note that the results obtained in this section can be easily generalized to the case of uneven weights, which corresponds to the setting in agnostic federated learning (Mohri et al., 2019).

To demonstrate that increasing the number of clients $K$ can significantly reduce the generalization error of an FL algorithm in certain scenarios, Barnes et al. (2022) assumes a Bregman loss. Specifically, for a strictly convex function $f : \mathbb{R}^d \to \mathbb{R}$, the Bregman divergence between $x, y \in \mathbb{R}^d$ is defined as: $D_f(x, y) \triangleq f(x) - f(y) - \langle \nabla f(y), x - y \rangle$ (Bregman, 1967). A Bregman loss is then defined based on the Bregman divergence, i.e. $\ell(w, z) = D_f(w, z)$.

While Barnes et al. (2022) proves that $|\mathcal{E}_{OG}(\mathcal{A})|$ decays with a fast rate with respect to $K$, the following result shows that under a Bregman loss, the overall generalization error $|\mathcal{E}_{\mathcal{D}}(\mathcal{A})|$ for an FL algorithm using model averaging achieves such a fast convergence rate with respect to $K$, namely both $|\mathcal{E}_{OG}(\mathcal{A})|$ and $|\mathcal{E}_{PG}(\mathcal{A})|$ exhibit this fast rate.

**Theorem 5.1.** *Let $\ell(w, z) = D_f(w, z)$, and assume that*

*(i) $(-1)^{V_i'}\left(\ell(W_i, \tilde{z}_{1,\bar{u}_1^{i,1}}^{i,1}) - \ell(W_i, \tilde{z}_{1,\bar{u}_1^{i,0}}^{i,0})\right)$ is $\sigma_i^2$-subGaussian under $P_{V_i'} \otimes P_{W_i|\tilde{z}^i,u^i}$ for any $i$,*

*(ii)* $(-1)^{U'^{i,v_i}_j} \left( \ell(W_i, \tilde{z}^{i,v_i}_{j,1}) - \ell(W_i, \tilde{z}^{i,v_i}_{j,0}) \right)$ *is* $\tilde{\sigma}^2_{i,j}$*-sub-Gaussian under* $P_{U'^{i,v_i}_j | v_i} \otimes P_{W_i | \tilde{z}^i, v_i}$ *for any* $i, j$.

*Then, for* $\mathcal{A}$ *using model averaging,*

$$|\mathcal{E}_\mathcal{D}(\mathcal{A})| \leq \frac{1}{K^2} \sum_{i=1}^K \mathbb{E}_{\tilde{Z}^i, U^i} \sqrt{2\sigma_i^2 I^{\tilde{Z}^i, U^i}(W_i; V_i)}$$

$$+ \frac{1}{K^2 n} \sum_{i=1}^K \sum_{j=1}^n \mathbb{E}_{\tilde{Z}^i, V_i} \sqrt{2\sigma^2_{i,j} I^{\tilde{Z}^i, V_i}(W_i; U^{i,V_i}_j)}.$$

**Remark 5.1.** *The assumptions given in (i-ii) are weaker than the boundedness assumption for the loss function, as if* $\ell$ *is bounded, then the difference between two losses is also bounded, which is guaranteed to be sub-Gaussian. In the case of heavy-tailed losses, additional techniques such as truncation (Dong et al., 2024) would be required to ensure the results hold. Furthermore, Theorem 5.1 can be interpreted as an end-to-end generalization bound for a single round of FedAvg. Extending it to a multi-round setting is straightforward by following the same approach used in Barnes et al. (2022, Corollary 3).*

Compared to the results in Section 3, Theorem 5.1 introduces two notable distinctions. First, the global CMI terms (i.e., CMI based on $W$) are replaced by local CMI terms (i.e., CMI based on $W_i$). Second, an additional $\frac{1}{K}$ factor appears in the bound, resulting in a faster convergence rate. While the data-processing inequality indicates that global CMI terms are smaller than local CMI terms, the specific model aggregation strategy and loss function allow us to explicitly observe this fast-rate behavior.

Particularly, in this setting, we rely solely on local privacy constraints to guarantee generalization. For example, if each $\mathcal{A}_i$ is $\epsilon$-differentially private for all $i$, then we obtain $|\mathcal{E}_\mathcal{D}(\mathcal{A})| \leq \mathcal{O}\left( \frac{1}{K}\sqrt{\frac{\epsilon}{K}} + \frac{1}{K}\sqrt{\frac{\epsilon}{n}} \right)$, where the derivation is nearly the same to Lemma 3.2, and the maximum value of the sub-Gaussian variance proxies is considered. This result clearly improves upon the bounds in Section 3 (cf. Remark 3.2). Additionally, Barnes et al. (2022) also presents a *communication constraint*-based result, which can also be applied to the overall FL generalization error. Specifically, if each client $i$ can only transmit $B$ bits of information to the central server, then each $W_i$ can take at most $2^B$ distinct values after quantization. In this case, $I^{\tilde{Z}^i, U^i}(W_i; V_i) \leq H(W_i) \leq B\log 2$ and $I(W_i; U^{i,V_i}|\tilde{Z}^i, V_i) \leq B\log 2$. As a result, the generalization error is bounded by $|\mathcal{E}_\mathcal{D}(\mathcal{A})| \leq \mathcal{O}\left( \frac{\sqrt{B}}{K} + \frac{1}{K}\sqrt{\frac{B}{n}} \right)$.

Although many popular loss functions, such as the squared loss and KL divergence, can be regarded as special cases of Bregman divergence (Yamane et al., 2023), the Bregman loss $D_f(w, z)$ used above requires $w$ and $z$ to have the same dimension. This requirement may limit its applicabil-

ity beyond nonparametric algorithms. Recently, Gholami & Seferoglu (2024) achieves an even faster decay rate for $|\mathcal{E}_{OG}(\mathcal{A})|$ by using a smooth and strongly convex loss function. Building on their assumptions and techniques, we extend this analysis in an information-theoretic framework. Additionally, we assume that each participating client's local algorithm is an interpolating algorithm (i.e., achieving zero empirical risk). This leads to the following result.

**Theorem 5.2.** *Let* $\ell(\cdot, \cdot) \in [0, 1]$ *be* $L$*-smooth and* $\alpha$*-strongly convex in* $\mathcal{W}$*, and let* $\mathcal{A}_i$ *be an interpolating algorithm for each client* $i$. *Let* $\gamma = \frac{2L}{\alpha \log 2}$*, then*

$$|\mathcal{E}_{OG}(\mathcal{A})| \leq \frac{\gamma}{K^3 n} \sum_{i=1}^K \sum_{j=1}^n I(W_i; U^{i,V_i}_j | \tilde{Z}^i, V_i)$$

$$+ \frac{2\sqrt{\gamma}}{K^2 n} \sum_{i=1}^K \sum_{j=1}^n \sqrt{\mathbb{E}\left[ \ell(W, Z_{i,j}) \right] I(W_i; U^{i,V_i}_j | \tilde{Z}^i, V_i)}.$$

Clearly, the decay rate with respect to $K$ in this bound is even faster than that for $|\mathcal{E}_{OG}(\mathcal{A})|$ in Theorem 5.1. Note that the boundedness assumption can be relax to sub-Gaussianity (i.e., an "almost bounded" case). The assumption of an interpolating algorithm is also practical due to the widespread use of overparameterized deep neural networks in modern deep learning scenarios.

Regarding the local CMI term $I(W_i; U^{i,V_i}_j | \tilde{Z}^i, V_i)$ in Theorem 5.2, note that in our single-round setting, the local model is trained independently and does not communicate with other clients. As a result, the CMI term associated with client $i$ is, by construction, independent of the CMI terms of other clients. It reflects only the relationship between a single client's data and its corresponding model and therefore does not, in general, provide a direct or definitive measure of cross-client data heterogeneity. However, if the data distribution itself is governed by a heterogeneity parameter, as in the experimental setup of Sun et al. (2024), then the value of this CMI term may indeed vary with that parameter, as it ultimately depends on the underlying data distribution.

Moreover, the use of smooth and strongly convex loss functions, often employed to ensure the linear convergence of gradient descent (GD), has also been studied in the context of personalized FL generalization. For example, Chen et al. (2021) shows that if the data heterogeneity—measured as the minimum average (squared) $L_2$ distance between global model parameters and optimal local parameters—is mild, then FedAvg achieves minimax optimality. In Theorem 5.2, data heterogeneity is captured by the factor $\mathbb{E}\left[ \ell(W, Z_{i,j}) \right]$ in the second term of the bound. In fact, if $\mathcal{A}_i$ does not interpolate the training data, this factor is replaced by $\mathbb{E}\left[ \ell(W, Z_{i,j}) \right] - \mathbb{E}\left[ \ell(W_i, Z_{i,j}) \right]$. When this factor vanishes, i.e., when data heterogeneity is negligible, the second term (which contains the square-root function) becomes neg-

ligible, and the first term (with a faster rate, e.g., $\mathcal{O}\left(\frac{B}{K^2}\right)$ for a communication constraint) dominates the bound. This observation aligns with Chen et al. (2021), showing that FedAvg performs better when data heterogeneity is low.

Additionally, we remark that extending a similar analysis to bound $|\mathcal{E}_{PG}(\mathcal{A})|$ is not straightforward. This would require that, for each client, there exists a minimizer that achieves a zero-gradient signal for any unseen local data, which is generally not guaranteed.

## 6. Empirical Verification

In this section, we empirically evaluate the effectiveness of our CMI bounds by comparing them to the expected generalization gap. Specifically, we estimate the fast-rate e-CMI bound for FL, as presented in Theorem 4.1, referred to as the *Fast-rate Bound* in our experiments. Additionally, due to the challenges associated with estimating MI when dealing with high-dimensional random variables, we compute an evaluated version of the CMI bound from Theorem 3.1. This e-CMI bound, derived in Theorem G.1 (in Appendix), serves as a lower bound for Theorem 3.1 and is denoted as the *Square-root Bound* in our experiments. Our experimental setup, particularly the construction of supersamples, closely follows previous works (Harutyunyan et al., 2021; Hellström & Durisi, 2022a; Wang & Mao, 2023c; 2024b; Dong et al., 2024), and we construct the superclient in a similar manner. For the FL learning process, we implement FedAvg following the federated training protocol of McMahan et al. (2017), with modifications to reduce computational overhead. We conduct image classification experiments on two datasets: MNIST (LeCun et al., 2010) and CIFAR-10 (Krizhevsky, 2009). The details of our experimental setup and results are presented below.

### 6.1. Experiment Setup

For the MNIST digit recognition task, we train a CNN with approximately 170K parameters. Each local training algorithm $\mathcal{A}_i$ trains this model using GD. For CIFAR-10, we use the same CNN model from McMahan et al. (2017), which has approximately $10^6$ parameters. As also discussed in McMahan et al. (2017), the SOTA models, such as vision transformers (ViTs) (Dosovitskiy et al., 2021), achieve over 99% accuracy on CIFAR-10. However, since our focus is on verifying generalization bounds rather than achieving SOTA performance, training a ViT-scale model would be unnecessarily costly. Therefore, a simple CNN suffices for our purpose. Each local training algorithm $\mathcal{A}_i$ trains the CNN model using mini-batch SGD.

For both of the classification tasks, clients train locally for five epochs per round before sending their models to the central server, with training spanning 300 communication

rounds (which reduces the commonly used 1000 rounds to lower computational costs). Additionally, we apply a pathological non-IID data partitioning scheme as in McMahan et al. (2017): data are sorted by label, split into 200 shards of size 300, and each client is randomly assigned 2 shards, and we evaluate prediction error as our performance metric, i.e. we utilize the zero-one loss function to compute generalization error. During training, we use cross-entropy loss as a surrogate to enable optimization with gradient-based methods. Since we pre-define the non-participating clients from the superclient, all participating clients are included in every round of training.

The complete experiment details including other hyperparameter settings and CMI estimation can be found in Appendix H.

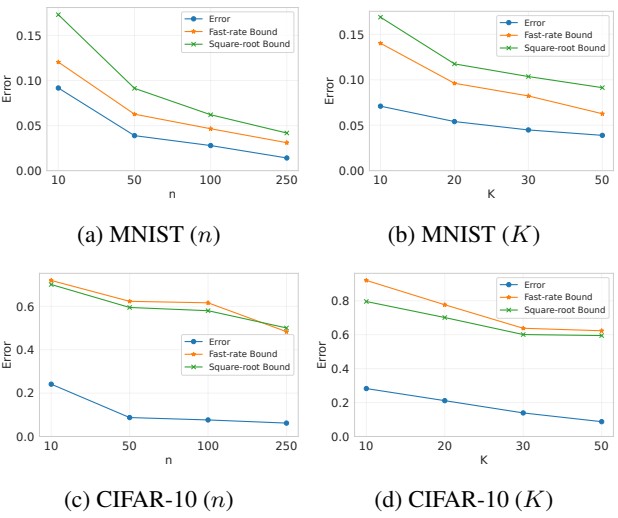

(a) MNIST ($n$)      (b) MNIST ($K$)

(c) CIFAR-10 ($n$)      (d) CIFAR-10 ($K$)

Figure 1: Verification of bounds on MNIST and CIFAR-10. (a-b) Results on MNIST with different $n$ and $K$. (c-d) Results on CIFAR-10 with different $n$ and $K$.

### 6.2. Results

Figure 1 presents the estimated expected generalization gap alongside our CMI bounds. Notably, both CMI bounds are non-vacuous, meaning they remain below the upper bound of the loss function (i.e., 1). More importantly, they effectively capture the generalization behavior of the FL learning algorithm and are numerically very close to the expected generalization gap. For MNIST, where the algorithm achieves a small training error, the fast-rate bound from Theorem 4.1 performs significantly better than the square-root CMI bound in Theorem 3.1 in this low empirical risk regime. On CIFAR-10, since we use a simple CNN model and limit the number of communication rounds, the empirical risk remains non-negligible. In this case, we observe that the evaluated version of Theorem 3.1 outperforms Theorem 4.1

in most settings. While the tightness of e-CMI bounds in standard centralized i.i.d. learning has been observed in previous works (Harutyunyan et al., 2021; Hellström & Durisi, 2022a; Wang & Mao, 2023c; Dong et al., 2024), our results confirm that CMI bounds remain numerically tight in FL.

## 7. Other Related Literature

**Federated Learning Methods** FedAvg (McMahan et al., 2017) is one of the earliest and most widely used FL algorithms. However, its performance degrades significantly in non-i.i.d. settings. To mitigate this issue, FedProx (Li et al., 2020) introduces a regularization term, $\|w - w_i\|$, which helps stabilize updates and reduce divergence among local models. In fact, the squared form of this regularization term naturally arises in the derivation of our Theorem 5.2 (see Appendix F.2), and when the CMI term in Theorem 5.2 is small, this regularization term also tends to be small. Another approach, SCAFFOLD (Karimireddy et al., 2020), tackles data heterogeneity using variance reduction techniques to improve convergence. FedOpt (Reddi et al., 2021) extends adaptive optimization methods to FL, enabling more efficient learning dynamics, while FedNova (Wang et al., 2020) introduces a normalized averaging scheme to mitigate objective inconsistencies while preserving fast training convergence. More recently, leveraging information-theoretic analysis, FedMDMI (Zhang et al., 2024) incorporates a global model-data MI term as a regularizer to improve generalization performance. Beyond these methods, numerous other FL algorithms have been proposed, we refer readers to (Huang et al., 2024; Solans et al., 2024) for recent advances.

**Generalization Theory in Federated Learning** Recent research has increasingly focused on understanding generalization in FL. In addition to works in (Chen et al., 2021; Hu et al., 2023; Yagli et al., 2020; Barnes et al., 2022; Gholami & Seferoglu, 2024; Zhang et al., 2024), which we have discussed throughout the paper, several other studies provide new insights. Sefidgaran et al. (2022; 2024) introduce rate-distortion-based generalization bounds for FL, with Sefidgaran et al. (2024) demonstrating that increasing the frequency of communication rounds does not always reduce generalization error and, in some cases, may even degrade generalization performance. Stability-based generalization analysis has also been explored in FL. For instance, Sun et al. (2024) analyze the impact of data heterogeneity, concluding that greater heterogeneity leads to higher generalization error. It is also worth noting that generalization theory in meta-learning is closely related to FL. Particularly, personalized FL can be viewed as a variant of meta-learning (Fallah et al., 2020). In fact, our CMI framework is inspired in part by a recent CMI-based generalization bound for meta-learning proposed in Hellström & Durisi (2022b). However, our results are not directly comparable to theirs due to a key difference in problem setup: their meta-learning framework requires the meta-learner (i.e., the global model $W$) to be further trained on the test tasks (i.e., previously non-participating clients), whereas in FL, the global model is evaluated directly on unseen clients without any additional local fine-tuning. To enable a direct comparison with Hellström & Durisi (2022b), our CMI framework would need to be extended to the personalized FL setting, where local adaptation of the global model is permitted.

## 8. Concluding Remarks

In this paper, we present novel CMI bounds for FL in the two-level generalization setting. Empirical results show that our e-CMI bounds are numerically non-vacuous. Future research directions include exploring the relationship between model aggregation frequency and local updates through CMI analysis, and extending our framework to accommodate unbounded loss functions beyond the sub-Gaussianity.

## Acknowledgements

The authors would like to thank the anonymous AC and reviewers for their careful reading and valuable suggestions. The authors acknowledge the use of GPT-4o to assist in polishing the language of this work. All AI-generated text was carefully reviewed and edited by the authors, who take full responsibility for the final content of the paper.

## Impact Statement

The primary focus of our work is on advancing the theoretical foundations of federated learning, aiming to deepen our understanding of this field. As with most theoretical work, it inherently presents challenges in explicitly highlighting potential societal consequences. Meanwhile, the proposed theoretical framework, particularly in privacy constraint based results, may find application in related research domains like trustworthy AI. While the theoretical results in such domains could have positive societal impacts, it's crucial to acknowledge the potential for misuse. The research community plays a vital role in delivering guidance on responsible and ethical applications to navigate the evolving landscape of technology responsibly.

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

# Appendices

The structure of the Appendix is outlined as follows. The appendix begins with begins with a figure illustrating the superclient and supersample construction, followed by a table summarizing the key notations used throughout the paper. Section B presents a collection of technical lemmas essential for our analysis. In Section C, we provide IOMI generalization bounds for FL. Sections D, E, and F contain detailed proofs of our theoretical results. Additionally, an extra eCMI result is presented in Section G. For further details regarding the experimental setup, refer to Section H.

## A. Visualization of CMI Framework in FL and Notation Summary

Figure 2 illustrates the superclient and supersample construction used in our CMI framework for FL. For clarity and ease of reference, Table 1 compiles the key notations used throughout the paper.

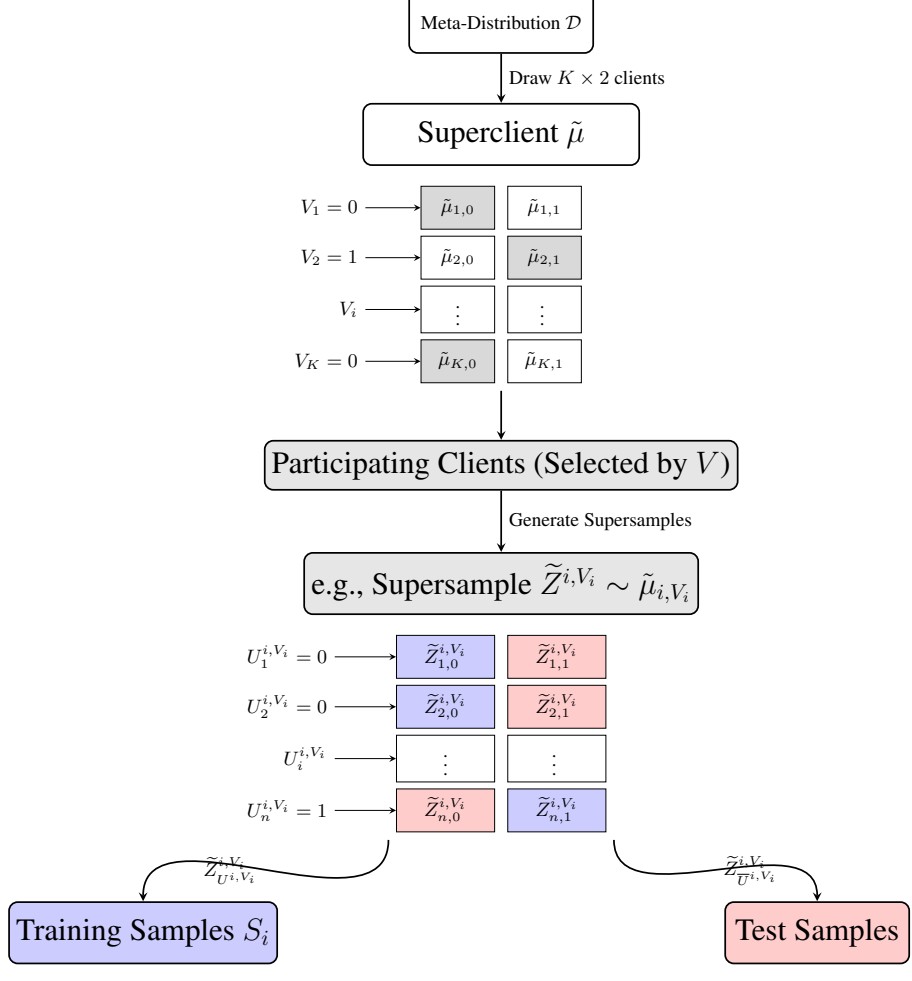

Figure 2: Illustration of the Superclient and Supersample Construction in FL

## B. Technical Lemmas and Additional Definitions

The following lemmas are widely used in our paper.

**Lemma B.1** (Donsker-Varadhan variational representation of KL divergence (Donsker & Varadhan, 1983)). *Let $Q$, $P$ be probability measures on $\Theta$, for any bounded measurable function $f : \Theta \to \mathbb{R}$, we have $\mathrm{D}_{\mathrm{KL}}(Q||P) = \sup_f \mathbb{E}_{\theta \sim Q}[f(\theta)] - \ln \mathbb{E}_{\theta \sim P}[\exp f(\theta)]$.*

Table 1: Summary of Notations

| Notation | Description |
|---|---|
| $\mathcal{Z}, \mathcal{W}, \mathcal{C}$ | Instance space, hypothesis space, and set of all possible clients |
| $\mathcal{D}$ | Meta-distribution over clients |
| $\mu_i$ | Local data distribution for client $i$ |
| $K, n$ | Number of participating clients, dataset size per client |
| $S_i, S$ | Training dataset of client $i$, union of all training datasets |
| $\mathcal{A}_i, \mathcal{A}$ | Local learning algorithm, global FL algorithm |
| $P_{W_i|S_i}, P_{W|S}$ | Conditional distributions of local and global models given training data |
| $L_{\mathcal{D}}(W)$ | Global true risk (population risk) |
| $L_{\mu_i}(W)$ | True risk of participating client $i$ |
| $L_{S_i}(W), L_S(W)$ | Empirical risk of client $i$, average empirical risk for all participating clients |
| $\mathcal{E}_{\mathcal{D}}(\mathcal{A})$ | Expected generalization error |
| $\mathcal{E}_{PG}(\mathcal{A})$ | Participation gap |
| $\mathcal{E}_{OG}(\mathcal{A})$ | Out-of-sample gap |
| $\mathcal{E}_{ER}(\mathcal{A})$ | Expected excess risk |
| $\tilde{\mu}$ | Superclient matrix drawn from meta-distribution $\mathcal{D}$ |
| $\widetilde{Z}^{i,b}$ | Supersample for client distribution $\tilde{\mu}_{i,b}$ where $b \in \{0,1\}$ |
| $\widetilde{Z}^i$ | $(\widetilde{Z}^{i,0}, \widetilde{Z}^{i,1})$ |
| $\widetilde{Z}$ | Collection of all supersamples $\{\widetilde{Z}^i\}_{i=1}^K$ |
| $V_i$ | Bernoulli variable determining client participation in $i$-th row of $\tilde{\mu}$ |
| $U_j^{i,b}$ | Bernoulli variable determining training data in $j$-th row of supersample $\widetilde{Z}^{i,b}$ |
| $V = \{V_i\}_{i=1}^K$ | Bernoulli random variables indicating client participation |
| $U^i$ | $(U^{i,0}, U^{i,1})$ |
| $U = \{U^i\}_{i=1}^K$ | Bernoulli random variables determining sample usage in supersamples |
| $\overline{V}_i, \overline{U}_j^{i,b}$ | Complementary Bernoulli variables for participation and sample selection |
| $\widetilde{Z}^{i,b}_{U^{i,b}}, \widetilde{Z}^{i,b}_{\overline{U}^{i,b}}$ | Training and test samples from supersample drawn from $i$-th row, $b$-th column of $\tilde{\mu}$ |
| $\widetilde{Z}^{i,b}_{j,U_j^{i,b}}, \widetilde{Z}^{i,b}_{j,\overline{U}_j^{i,b}}$ | The j-th elements in training and test samples from $i$-th row, $b$-th column of $\tilde{\mu}$ |
| $W$ | Global model aggregated from local models $\{W_i\}_{i=1}^K$ |
| $W_i$ | Local model produced by client $i$ |
| $w^*$ | Global risk minimizer under $\mathcal{D}$ |

**Lemma B.2** (Hoeffding's Lemma (Hoeffding, 1963)). *Let $X \in [a, b]$ be a bounded random variable with mean $\mu$. Then, for all $t \in \mathbb{R}$, we have $\mathbb{E}\left[e^{tX}\right] \leq e^{t\mu + \frac{t^2(b-a)^2}{8}}$.*

**Lemma B.3** (Hoeffding's inequality (Hoeffding, 1963)). *Let $X_1, \ldots, X_m$ be independent random variables with $X_i$ taking values in $[a_i, b_i]$ for all $i \in [m]$. Then, for any $\epsilon > 0$, the following inequalities hold for $S_m = \sum_{i=1}^m X_i$:*

$$\Pr\left(|S_m - \mathbb{E}[S_m]| \geq \epsilon\right) \leq 2e^{-\frac{-2\epsilon^2}{\sum_{i=1}^m (b_i - a_i)^2}}.$$

**Information-Theoretic Generalization Bound** The original information-theoretic bound in Xu & Raginsky (2017) is for centralized learning, or a single data distribution equivalently, and the key component in the bound is the mutual information between the output $W$ and the entire input sample $S$. This result is given as follows:

**Lemma B.4** (Xu & Raginsky (2017, Theorem 1.)). *Let the size of training dataset $S$ be $n$. Assume the loss $\ell(w, Z)$ is $\sigma$-sub-Gaussian[2] for any $w \in \mathcal{W}$, then the generalization error for the learning algorithm $\mathcal{A}$ is upper bounded by*

$$|\mathcal{E}(\mathcal{A})| \leq \sqrt{\frac{2\sigma^2}{n} I(W; S)}.$$

---

[2]A random variable $X$ is $\sigma$-sub-Gaussian if for any $\lambda \in \mathbb{R}$, $\log \mathbb{E} \exp(\lambda(X - \mathbb{E}X)) \leq \lambda^2 \sigma^2/2$. Note that a bounded loss is guaranteed to be sub-Gaussian.

**Definition B.1** (Differential Privacy (Dwork et al., 2006b;a)). *An algorithm $\mathcal{A} : \mathcal{Z}^n \to \mathcal{W}$ is $(\epsilon, \delta)$-differentially private (DP) if, for any two data sets $s, s' \in \mathcal{Z}^n$ that differ in a single element and for any measurable set $W \subset \mathcal{W}$,*

$$\Pr[A(s) \in W] \leq e^\epsilon \cdot \Pr[A(s') \in W] + \delta.$$

*If $\delta = 0$, $\mathcal{A}$ is $\epsilon$-differentially private.*

## C. Input-Output Mutual Information (IOMI) Generalization Bound

In fact, the IOMI bound in Lemma B.4 can be directly applied to the FL setting, leading to the following bound on $\mathcal{E}_{OG}(\mathcal{A})$.

**Theorem C.1.** *For each $i \in [K]$, assume the loss $\ell(w, Z)$ is $\sigma_i$-subGaussian with respect to $Z \sim \mu_i$ for any $w \in \mathcal{W}$. Let $\sigma_{\max} = \max\{\sigma_i\}_{i=1}^K$, then the out-of-sample gap of FL learning algorithm $\mathcal{A}$ is upper bounded by*

$$|\mathcal{E}_{OG}(\mathcal{A})| \leq \sqrt{\frac{2\sigma_{\max}^2}{nK} I(W; S_{1:K})}. \tag{2}$$

Clearly, increasing the amount of data per client improves generalization performance. More importantly, the key term $I(W; S_{1:K})$ in the bound suggests that if the hypothesis $W$ contains minimal information about the training data $S_{1:K}$ of the clients (i.e. $I(W; S_{1:K})$ is small), then the FL algorithm $\mathcal{A}$ will exhibit a small out-of-sample gap. This aligns with a desirable property of federated learning: good client privacy implies good generalization. This relationship is also discussed in Lemma 3.2.

To further tighten the generalization bound for $\mathcal{E}_{OG}(\mathcal{A})$, we adopt techniques introduced by Bu et al. (2020), along with similar developments from Wang & Mao (2023b) for bounding $\mathcal{E}_{PG}(\mathcal{A})$.

**Theorem C.2.** *For each $i \in [K]$, assume the loss $\ell(w, Z)$ is $\sigma_i$-subGaussian under any $\mu \sim \mathcal{D}$ for all $w \in \mathcal{W}$, then the generalization error of FL learning algorithm $\mathcal{A}$ is upper bounded by*

$$|\mathcal{E}_{\mathcal{D}}(\mathcal{A})| \leq \frac{1}{K} \sum_{i=1}^K \sqrt{2\sigma^2 \mathrm{D}_{\mathrm{KL}}(P_Z || \mu_i)} + \frac{1}{nK} \sum_{i=1}^K \sum_{j=1}^n \sqrt{2\sigma^2 I(W; Z_{i,j})}, \tag{3}$$

*where $P_Z$ is the data distribution induced by the meta-distribution, e.g., $P_Z(Z = z) = \int_\mu \mu(Z = z) d\mathcal{D}(\mu)$.*

The proof of this theorem follows directly from the techniques in Bu et al. (2020, Proposition 1) and Wang & Mao (2023b, Theorem 4.1).

Compared to the CMI bounds, this bound has its own distinctive advantage: the first term explicitly characterizes how the divergence between client distributions affects the participation gap. This provides a clearer understanding of how heterogeneity in client distributions impacts the generalization performance of FL algorithms. In addition, this result will ultimately lead to a gradient-norm-based regularizer when SGD or SGLD is used (Wang & Mao, 2024a) and hints at a potential feature alignment mechanism in the KL sense for clients.

## D. Omitted Proofs in Section 3

### D.1. Proof of Lemma 3.1

*Proof.* First, due to the symmetric property of superclient construction, we have

$$\mathcal{E}_{PG}(\mathcal{A}) = \mathbb{E}_W \left[ L_\mathcal{D}(W) \right] - \mathbb{E}_{W, \mu_{[K]}} \left[ L_{\mu_{[K]}}(W) \right]$$

$$= \mathbb{E}_{W, \mu, Z} \left[ \ell(W, Z) \right] - \frac{1}{K} \sum_{i=1}^K \mathbb{E}_{W, \mu_i, Z_i'} \left[ \ell(W, Z_i') \right]$$

$$= \frac{1}{K} \sum_{i=1}^K \mathbb{E}_{\tilde{\mu}, W, V_i} \left[ \left( L_{\tilde{\mu}_{i,\overline{V}_i}}(W) - L_{\tilde{\mu}_{i,V_i}}(W) \right) \right]$$

$$= \frac{1}{K} \sum_{i=1}^K \mathbb{E}_{\tilde{\mu}, W, V_i} \left[ (-1)^{V_i} \left( L_{\tilde{\mu}_{i,1}}(W) - L_{\tilde{\mu}_{i,0}}(W) \right) \right]. \tag{4}$$

For example, if $V_i = 0$, then $L_{\tilde{\mu}_{i,1}}(W)$ is the testing error of non-participating client $\tilde{\mu}_{i,1}$, and $L_{\tilde{\mu}_{i,0}}(W)$ is the local testing error of participating client $\tilde{\mu}_{i,0}$. In addition, $V_i = 1$ will switch the roles of clients $\tilde{\mu}_{i,1}$ and $\tilde{\mu}_{i,0}$, and still reflect the gap of testing errors between non-participating client and participating client. In other words, $\frac{1}{K}\sum_{i=1}^{K}(-1)^{V_i}\left(L_{\tilde{\mu}_{i,1}}(W) - L_{\tilde{\mu}_{i,0}}(W)\right)$ in Eq. (4) is an unbiased estimator of $\mathcal{E}_{PG}(\mathcal{A})$.

Furthermore, notice that, by definition, we have

$$
\begin{aligned}
\mathbb{E}_{\tilde{\mu},W,V_i}\left[(-1)^{V_i}L_{\tilde{\mu}_{i,1}}(W)\right] &= \mathbb{E}_{\tilde{\mu},W,V_i,Z\sim\tilde{\mu}_{i,1}}\left[(-1)^{V_i}\ell(W,Z)\right] \\
&= \mathbb{E}_{W,V_i,\widetilde{Z}^{i,1},U_j^{i,1}}\left[(-1)^{V_i}\ell(W,\widetilde{Z}_{j,\overline{U}_j^{i,1}}^{i,1})\right],
\end{aligned}
\tag{5}
$$

and

$$
\begin{aligned}
\mathbb{E}_{\tilde{\mu},W,V_i}\left[(-1)^{V_i}L_{\tilde{\mu}_{i,0}}(W)\right] &= \mathbb{E}_{\tilde{\mu},W,V_i,Z\sim\tilde{\mu}_{i,0}}\left[(-1)^{V_i}\ell(W,Z)\right] \\
&= \mathbb{E}_{W,V_i,\widetilde{Z}^{i,0},U_j^{i,0}}\left[(-1)^{V_i}\ell(W,\widetilde{Z}_{j,\overline{U}_j^{i,0}}^{i,0})\right],
\end{aligned}
\tag{6}
$$

where $j$ in both equations can be arbitrary index from $[n]$. We will simply let $j = 1$ from now on.

Note that the expectation over $\tilde{\mu}$ does not need to explicitly appear in Eq. (5-6) as evaluating function $\ell$ does not rely on the random variable $\tilde{\mu}$.

Consequently, plugging Eq. (5-6) into Eq. (4) will give us the desired result:

$$
\mathcal{E}_{PG}(\mathcal{A}) = \frac{1}{K}\sum_{i=1}^{K}\mathbb{E}_{\widetilde{Z}^i,U^i,W,V_i}\left[(-1)^{V_i}\left(\ell(W,\widetilde{Z}_{1,\overline{U}_1^{i,1}}^{i,1}) - \ell(W,\widetilde{Z}_{1,\overline{U}_1^{i,0}}^{i,0})\right)\right].
$$

Then similarly, according to the symmetric property of supersample construction, we have

$$
\begin{aligned}
\mathcal{E}_{OG}(\mathcal{A}) &= \mathbb{E}_{W,\mu_{[K]}}\left[L_{\mu_{[K]}}(W)\right] - \mathbb{E}_{W,S}\left[L_S(W)\right] \\
&= \frac{1}{K}\sum_{i=1}^{K}\mathbb{E}_{W,\mu_i,Z_i'}\left[\ell(W,Z_i')\right] - \frac{1}{Kn}\sum_{i=1}^{K}\sum_{j=1}^{n}\mathbb{E}_{W,\mu_i,Z_{i,j}}\left[\ell(W,Z_{i,j})\right] \\
&= \frac{1}{Kn}\sum_{i=1}^{K}\sum_{j=1}^{n}\mathbb{E}_{\widetilde{Z}^i,W,V_i,U_j^{i,V_i}}\left[(-1)^{U_j^{i,V_i}}\left(\ell(W,\widetilde{Z}_{j,1}^{i,V_i}) - \ell(W,\widetilde{Z}_{j,0}^{i,V_i})\right)\right].
\end{aligned}
\tag{7}
$$

This completes the proof. $\qquad\square$

### D.2. Proof of Theorem 3.1

*Proof.* We start from bounding the participation gap.

Let $g(\tilde{z}^i,u^i,w,v_i) = (-1)^{v_i}\left(\ell(w,\tilde{z}_{1,u_1^{i,1}}^{i,1}) - \ell(w,\tilde{z}_{1,u_1^{i,0}}^{i,0})\right)$. To apply Lemma B.1, we let $f = t\cdot g$ for $t > 0$ in Lemma B.1, and let $V_i'$ be an independent copy of $V_i$ (i.e. $P_{V_i'} = P_{V_i}$ and $V_i' \perp\!\!\!\perp W|\tilde{z}^i,u^i$), then

$$
\mathbb{E}_{W,V_i|\tilde{z}^i,u^i}\left[g(\tilde{z}^i,u^i,W,V_i)\right] \le \inf_{t>0}\frac{I(W;V_i|\widetilde{Z}^i = \tilde{z}^i, U^i = u^i) + \log\mathbb{E}_{W,V_i'|\tilde{z}^i,u^i}\left[e^{tg(\tilde{z}^i,u^i,W,V_i')}\right]}{t}.
\tag{8}
$$

Since $V_i'$ is independent of $W, \widetilde{Z}^i$ and $U^i$, we have

$$
\mathbb{E}_{V_i'}\left[g(\tilde{z}^i,u^i,w,V_i')\right] = \mathbb{E}_{V_i'}\left[(-1)^{V_i'}\left(\ell(w,\tilde{z}_{1,u_1^{i,1}}^{i,1}) - \ell(w,\tilde{z}_{1,u_1^{i,0}}^{i,0})\right)\right] = 0
$$

for any $w, \tilde{z}^i$ and $u^i$. Ergo,

$$
\mathbb{E}_{W|\tilde{z}^i,u^i}\left[\mathbb{E}_{V_i'}\left[g(\tilde{z}^i,u^i,W,V_i')\right]\right] = 0.
$$

In addition, as the loss is bounded in $[0, 1]$, we know that $g(\cdot, \cdot, \cdot, \cdot) \in [-1, 1]$.

Thus, $g(\tilde{z}^i, u^i, W, V_i')$ is a zero-mean random variable bounded in $[-1, 1]$ for a fixed pair of $(\tilde{z}^i, u^i)$. By Lemma B.2, we have

$$\mathbb{E}_{W, V_i' | \tilde{z}^i, u^i} \left[ e^{tg(\tilde{z}^i, u^i, W, V_i')} \right] \le e^{\frac{t^2}{2}}. \tag{9}$$

Plugging the above into Eq. (8),

$$\mathbb{E}_{W, V_i | \tilde{z}^i, u^i} \left[ g(\tilde{z}^i, u^i, W, V_i) \right] \le \inf_{t > 0} \frac{I(W; V_i | \widetilde{Z}^i = \tilde{z}^i, U^i = u^i) + \frac{t^2}{2}}{t} \tag{10}$$
$$= \sqrt{2I(W; V_i | \widetilde{Z}^i = \tilde{z}^i, U^i = u^i)},$$

where we choose $t = \sqrt{2I(W; V_i | \widetilde{Z}^i = \tilde{z}^i, U^i = u^i)}$ in the last equation.

Recall the formulation of $\mathcal{E}_{PG}(\mathcal{A})$ in Lemma 3.1 and by Jensen's inequality for the absolute value function, the upper bound for participation gap is obtained:

$$|\mathcal{E}_{PG}(\mathcal{A})| \le \frac{1}{K} \sum_{i=1}^{K} \mathbb{E}_{\widetilde{Z}^i, U^i} \left| \mathbb{E}_{W, V_i | \widetilde{Z}^i, U^i} \left[ (-1)^{V_i} \left( \ell(W, \widetilde{Z}_{1, U_1^{i,1}}^{i,1}) - \gamma(W, \widetilde{Z}_{1, U_1^{i,0}}^{i,0}) \right) \right] \right|$$
$$\le \frac{1}{K} \sum_{i=1}^{K} \mathbb{E}_{\widetilde{Z}^i, U^i} \sqrt{2I^{\widetilde{Z}^i, U^i}(W; V_i)}. \tag{11}$$

We then process similar procedure for the out-of-sample gap.

Let $h(\tilde{z}^i, v_i, w, u_j^{i, v_i}) = (-1)^{u_j^{i, v_i}} \left( \ell(w, \tilde{z}_{j,1}^{i, v_i}) - \ell(w, \tilde{z}_{j,0}^{i, v_i}) \right)$. To apply Lemma B.1, we let $f = t \cdot h$ for $t > 0$ in Lemma B.1, and let $U$ be an independent copy of $U$ (i.e. $P_U = P_{U'}$ and $U' \perp\!\!\!\perp W | \tilde{z}^i, v_i$), then

$$\mathbb{E}_{W, U_j^{i, v_i} | \tilde{z}^i, v_i} \left[ h(\tilde{z}^i, v_i, W, U_j^{i, v_i}) \right] \le \inf_{t > 0} \frac{I(W, U_j^{i, v_i} | \widetilde{Z}^i = \tilde{z}^i, V_i = v_i) + \log \mathbb{E}_{W, U_j'^{i, v_i} | \tilde{z}^i, v_i} \left[ e^{th(\tilde{z}^i, v_i, W, U_j'^{i, v_i})} \right]}{t}. \tag{12}$$

Since $U'$ is independent of $W, \widetilde{Z}^i$ and $V_i$, we have

$$\mathbb{E}_{U_j'^{i, v_i}} \left[ h(\tilde{z}^i, v_i, w, U_j'^{i, v_i}) \right] = \mathbb{E}_{U_j'^{i, v_i}} \left[ (-1)^{U_j'^{i, v_i}} \left( \ell(w, \tilde{z}_{j,1}^{i, v_i}) - \ell(w, \tilde{z}_{j,0}^{i, v_i}) \right) \right] = 0$$

for any $w, \tilde{z}^i$ and $v_i$. Hence,

$$\mathbb{E}_{W | \tilde{z}^i, v_i} \left[ \mathbb{E}_{U_j'^{i, v_i}} \left[ h(\tilde{z}^i, v_i, W, U_j'^{i, v_i}) \right] \right] = 0.$$

Again, since the loss is bounded in $[0, 1]$, we know that $h(\cdot, \cdot, \cdot, \cdot) \in [-1, 1]$.

Thus, $h(\tilde{z}^i, v_i, W, U_j'^{i, v_i})$ is a zero-mean random variable bounded in $[-1, 1]$ for a fixed pair of $(\tilde{z}^i, v_i)$. By Lemma B.2, we have

$$\mathbb{E}_{W, U_j'^{i, v_i} | \tilde{z}^i, v_i} \left[ e^{th(\tilde{z}^i, v_i, W, U_j'^{i, v_i})} \right] \le e^{\frac{t^2}{2}}. \tag{13}$$

Plugging the above into Eq. (12),

$$\mathbb{E}_{W, U_j^{i, v_i} | \tilde{z}^i, v_i} \left[ h(\tilde{z}^i, v_i, W, U_j^{i, v_i}) \right] \le \inf_{t > 0} \frac{I(W, U_j^{i, v_i} | \widetilde{Z}^i = \tilde{z}^i, V_i = v_i) + \frac{t^2}{2}}{t} \tag{14}$$
$$= \sqrt{2I(W; U_j^{i, v_i} | \widetilde{Z}^i = \tilde{z}^i, V_i = v_i)},$$

where we choose $t = \sqrt{2I(W; U_j^{i,v_i} | \widetilde{Z}^i = \tilde{z}^i, V_i = v_i)}$ in the last equation.

Recall the formulation of $\mathcal{E}_{OG}(\mathcal{A})$ in Lemma 3.1 and by Jensen's inequality for the absolute value function, the upper bound for out-of-sample gap is obtained:

$$
\begin{aligned}
|\mathcal{E}_{OG}(\mathcal{A})| \leq & \frac{1}{Kn} \sum_{i=1}^{K} \sum_{j=1}^{n} \mathbb{E}_{\widetilde{Z}^i, V_i} \left| \mathbb{E}_{W, U_j^{i,V_i} | \widetilde{Z}^i, V_i} \left[ (-1)^{U_j^{i,V_i}} \left( \ell(W, \widetilde{Z}_{j,1}^{i,V_i}) - \ell(W, \widetilde{Z}_{j,0}^{i,V_i}) \right) \right] \right| \\
\leq & \frac{1}{Kn} \sum_{i=1}^{K} \sum_{j=1}^{n} \mathbb{E}_{\widetilde{Z}^i, V_i} \sqrt{2I^{\widetilde{Z}^i, V_i}(W; U_j^{i,V_i})}.
\end{aligned}
\tag{15}
$$

Combining Eq. (11) and Eq. (15), we have

$$
\begin{aligned}
|\mathcal{E}_{\mathcal{D}}(\mathcal{A})| \leq & |\mathcal{E}_{PG}(\mathcal{A})| + |\mathcal{E}_{OG}(\mathcal{A})| \\
\leq & \frac{1}{K} \sum_{i=1}^{K} \mathbb{E}_{\widetilde{Z}^i, U^i} \sqrt{2I^{\widetilde{Z}^i, U^i}(W; V_i)} + \frac{1}{Kn} \sum_{i=1}^{K} \sum_{j=1}^{n} \mathbb{E}_{\widetilde{Z}^i, V_i} \sqrt{2I^{\widetilde{Z}^i, V_i}(W; U_j^{i,V_i})}.
\end{aligned}
$$

This completes the proof. □

### D.3. Proof of Lemma 3.2

*Proof.* First, by the Jensen's inequality and the chain rule of MI, we have

$$
\frac{1}{K} \sum_{i=1}^{K} \mathbb{E}_{\widetilde{Z}^i, U^i} \sqrt{2I^{\widetilde{Z}^i, U^i}(W; V_i)} \leq \sqrt{\frac{2}{K} I(W; V | \widetilde{Z}, U)},
\tag{16}
$$

$$
\frac{1}{n} \sum_{j=1}^{n} \mathbb{E}_{\widetilde{Z}^i, V_i} \sqrt{2I^{\widetilde{Z}^i, V_i}(W; U_j^{i,V_i})} \leq \sqrt{\frac{2}{n} I(W; U^{i,V_i} | \widetilde{Z}^i, V_i)}.
\tag{17}
$$

The detailed application of Jensen's inequality and the chain rule of MI for obtaining the above inequalities are deferred to the proof for Corollary 3.1.

In FL, given $(\widetilde{Z}^i, V_i)$, $U^{i,V_i} - W_i - W$ forms a Markov chain. Thus, by the data-processing inequality, we have

$$
I(W; U^{i,V_i} | \widetilde{Z}^i, V_i) \leq I(W_i; U^{i,V_i} | \widetilde{Z}^i, V_i).
\tag{18}
$$

In addition, by the chain rule, we notice that

$$
I(W; V, \widetilde{Z}, U) = I(W; V | \widetilde{Z}, U) + I(W; \widetilde{Z}, U).
$$

Since the training dataset $S$ is determined by $(V, \widetilde{Z}, U)$, i.e. $S = \cup_{i=1}^{K} \widetilde{Z}_{U^{i,V}}^{i,V}$, we have $I(W; V, \widetilde{Z}, U) = I(W; S, V, \widetilde{Z}, U) = I(W; S)$. Then, due to the non-negativity of MI, $I(W; \widetilde{Z}, U) \geq 0$, we have

$$
I(W; V | \widetilde{Z}, U) \leq I(W; S).
\tag{19}
$$

Similarly, notice that

$$
I(W_i; U^{i,V_i}, \widetilde{Z}^i, V_i) = I(W_i; U^{i,V_i} | \widetilde{Z}^i, V_i) + I(W_i; \widetilde{Z}^i, V_i),
$$

Since the training dataset $S_i$ is determined by $(V_i, \widetilde{Z}^i, U^{i,V_i})$, i.e. $S_i = \widetilde{Z}_{U^{i,V}}^{i,V}$, we have $I(W_i; V_i, \widetilde{Z}^i, U^{i,V_i}) = I(W; S_i, V_i, \widetilde{Z}^i, U^{i,V_i}) = I(W; S_i)$. Then, due to the non-negativity of MI, $I(W_i; \widetilde{Z}^i, V_i) \geq 0$, we have

$$
I(W_i; U^{i,V_i} | \widetilde{Z}^i, V_i) \leq I(W_i; S_i).
\tag{20}
$$

Recall that each local algorithm $\mathcal{A}_i$ is $\epsilon_i$-differentially private, and the overall FL algorithm $\mathcal{A}$ is $\epsilon'$-differentially private,. By the definition of differential privacy (see Definition B.1), for any two local datasets $s_i$ and $s\prime_i$ differing by a single data point, and for any two aggregated datasets $s$ and $s\prime$ differing by a single data point, we have:

$$P(w_i|s_i) \leq e^{\epsilon_i} P(w_i|s'_i), \tag{21}$$

$$P(w|s) \leq e^{\epsilon'} P(w|s'). \tag{22}$$

The following development is similar to Barnes et al. (2022, Corollary 1).

$$
\begin{aligned}
I(W;S) &= \int_{w,s} dP(w,s) \log \frac{dP(w|s)}{dP(w)} \\
&= \int_{w,s} dP(w,s) \log \frac{dP(w|s)}{\mathbb{E}_{S'}[dP(w|S')]} \\
&\leq \int_{w,s} dP(w,s) \log \frac{dP(w|s)}{\inf_{s'} dP(w|s')} \\
&\leq \int_{w,s} dP(w,s)\epsilon' = \epsilon'.
\end{aligned}
$$

where the last inequality is by Eq. (22). On the other hand,

$$
\begin{aligned}
I(W;S) &\leq D_{\mathrm{KL}}(P_{W,S}||P_W P_S) + D_{\mathrm{KL}}(P_W P_S||P_{W,S}) \\
&= \int_{w,s} dP(w)dP(s) \left( \frac{dP(w|s)}{dP(w)} - 1 \right) \log \frac{dP(w|s)}{dP(w)} \\
&\leq \int_{w,s} dP(w)dP(s) \left( \frac{dP(w|s)}{\inf_{s'} dP(w|s')} - 1 \right) \log \frac{dP(w|s)}{\inf_{s'} dP(w|s')} \\
&\leq \int_{w,s} dP(w)dP(s) \left( e^{\epsilon'} - 1 \right) \epsilon' \\
&= \left( e^{\epsilon'} - 1 \right) \epsilon'.
\end{aligned}
$$

Hence,

$$I(W;S) \leq \min\{\epsilon', (e^{\epsilon'} - 1)\epsilon'\}. \tag{23}$$

The similar procedures can be applied to $I(W_i; S_i)$, which will give us

$$I(W_i; S_i) \leq \min\{\epsilon_i, (e^{\epsilon_i} - 1)\epsilon_i\}. \tag{24}$$

Putting Eq.(16-20) and Eq. (23-24) together, we have

$$\frac{1}{K} \sum_{i=1}^{K} \mathbb{E}_{\widetilde{Z}^i, U^i} \sqrt{I^{\widetilde{Z}^i, U^i}(W; V_i)} \leq \sqrt{\frac{\min\{\epsilon', (e^{\epsilon'} - 1)\epsilon'\}}{K}},$$

$$\frac{1}{n} \sum_{j=1}^{n} \mathbb{E}_{\widetilde{Z}^i, V_i} \sqrt{I^{\widetilde{Z}^i, V_i}(W; U_j^{i,V_i})} \leq \sqrt{\frac{\min\{\epsilon_i, (e^{\epsilon_i} - 1)\epsilon_i\}}{n}}.$$

This completes the proof. □

### D.4. Proof of Corollary 3.1

*Proof.* We first apply Jensen's inequality to the bound in Theorem 3.1,

$$\mathcal{E}_{\mathcal{D}}(\mathcal{A}) \leq \sqrt{\frac{2}{K} \sum_{i=1}^{K} I(W; V_i | \widetilde{Z}^i, U^i)} + \sqrt{\frac{2}{Kn} \sum_{i=1}^{K} \sum_{j=1}^{n} I(W; U_j^{i,V_i} | \widetilde{Z}^i, V_i)}. \tag{25}$$

For the first term in Eq. (25), notice that

$$I(W; V_i | \widetilde{Z}^i, U^i) = H(V_i) - H(V_i | W, \widetilde{Z}^i, U^i),$$
$$I(W; V_i | \widetilde{Z}, U) = H(V_i) - H(V_i | W, \widetilde{Z}, U).$$

Since conditioning reduces entropy, $H(V_i | W, \widetilde{Z}, U) \le H(V_i | W, \widetilde{Z}^i, U^i)$, we have $I(W; V_i | \widetilde{Z}^i, U^i) \le I(W; V_i | \widetilde{Z}, U)$.

Then, by the chain-rule of mutual information, we have

$$I(W; V | \widetilde{Z}, U) = \sum_{i=1}^{K} I(W; V_i | \widetilde{Z}, U, V_{[i-1]}).$$

Given that $V_i$ is i.i.d. sampled and is independent of $\widetilde{Z}$ and $U$, we know that $I(V_{[i-1]}; V_i | \widetilde{Z}, U) = 0$. Hence,

$$
\begin{aligned}
I(W; V | \widetilde{Z}, U) &= \sum_{i=1}^{K} I(W; V_i | \widetilde{Z}, U, V_{[i-1]}) + I(V_{[i-1]}; V_i | \widetilde{Z}, U) \\
&= \sum_{i=1}^{K} I(W, V_{[i-1]}; V_i | \widetilde{Z}, U) \\
&= \sum_{i=1}^{K} I(W; V_i | \widetilde{Z}, U) + I(V_{[i-1]}; V_i | \widetilde{Z}, U, W) \\
&\ge \sum_{i=1}^{K} I(W; V_i | \widetilde{Z}, U),
\end{aligned}
$$

where the last inequality is by the non-negativity of mutual information.

Consequently,

$$\sqrt{\frac{2}{K} \sum_{i=1}^{K} I(W; V_i | \widetilde{Z}^i, U^i)} \le \sqrt{\frac{2}{K} I(W; V | \widetilde{Z}, U)}. \tag{26}$$

We then focus on the second term in Eq. (25), notice that

$$I(W; U_j^{i,V_i} | \widetilde{Z}^i, V_i) = H(U_j^{i,V_i} | V_i) - H(U_j^{i,V_i} | W, \widetilde{Z}^i, V_i),$$
$$I(W; U_j^{i,V_i} | \widetilde{Z}, V) = H(U_j^{i,V_i} | V) - H(U_j^{i,V_i} | W, \widetilde{Z}, V).$$

Since $V_i$ is i.i.d. sampled and $U_j^{i,V_i}$ is independent of $V \setminus V_i$, i.e. the rest of $V_k$ for $k \ne i$, we have $H(U_j^{i,V_i} | V_i) = H(U_j^{i,V_i} | V)$. Moreover, according to conditioning reduces entropy, $H(U_j^{i,V_i} | W, \widetilde{Z}, V) \le H(U_j^{i,V_i} | W, \widetilde{Z}^i, V_i)$, we have $I(W; U_j^{i,V_i} | \widetilde{Z}^i, V_i) \le I(W; U_j^{i,V_i} | \widetilde{Z}, V)$.

Then, by the chain-rule of mutual information, we have

$$I(W; U^{i,V_i} | \widetilde{Z}, V) = \sum_{j=1}^{n} I(W; U_j^{i,V_i} | \widetilde{Z}, V, U_{[j-1]}^{i,V_i}).$$

For a given $V$, $U_j^{i,V_i}$ is i.i.d. sampled and is independent of $\widetilde{Z}$ and other $U_k^{i,V_i}$ for $k \ne i$, we know that

$I(U_{[j-1]}^{i,V_i}; U_j^{i,V_i} | \widetilde{Z}, V) = 0$. Hence,

$$
\begin{aligned}
I(W; U^{i,V_i} | \widetilde{Z}, V) &= \sum_{j=1}^{n} I(W; U_j^{i,V_i} | \widetilde{Z}, V, U_{[j-1]}^{i,V_i}) + I(U_{[j-1]}^{i,V_i}; U_j^{i,V_i} | \widetilde{Z}, V) \\
&= \sum_{j=1}^{n} I(W, U_{[j-1]}^{i,V_i}; U_j^{i,V_i} | \widetilde{Z}, V) \\
&= \sum_{j=1}^{n} I(W; U_j^{i,V_i} | \widetilde{Z}, V) + I(U_{[j-1]}^{i,V_i}; U_j^{i,V_i} | \widetilde{Z}, V, W) \\
&\geq \sum_{j=1}^{n} I(W; U_j^{i,V_i} | \widetilde{Z}, V),
\end{aligned}
$$

where the last inequality is by the non-negativity of mutual information.

Therefore, we have

$$
\sqrt{\frac{2}{Kn} \sum_{i=1}^{K} \sum_{j=1}^{n} I(W; U_j^{i,V_i} | \widetilde{Z}^i, V_i)} \leq \sqrt{\frac{2}{Kn} \sum_{i=1}^{K} I(W; U^{i,V_i} | \widetilde{Z}, V)}. \tag{27}
$$

Let $U^i = (U^{i,V_i}, U^{i,\overline{V}_i})$, then again by the chain-rule of mutual information, we have

$$
I(W; U | \widetilde{Z}, V) = \sum_{i=1}^{K} I(W; U^i | \widetilde{Z}, V, U^{[i-1]}) = \sum_{i=1}^{K} I(W; U^{i,V_i} | \widetilde{Z}, V, U^{[i-1]}),
$$

where the last equation is by the independence between $W$ and $U^{i,\overline{V}_i}$, namely $I(W; U^{i,\overline{V}_i} | \widetilde{Z}, V, U^{[i-1]}, U^{i,V_i}) = 0$.

Similarly, for a given $V$, $U^{i,V_i}$ is i.i.d. sampled and is independent of $\widetilde{Z}$ and other $U^k$ for $k \neq i$, we know that $I(U^{[i-1]}; U^{i,V_i} | \widetilde{Z}, V) = 0$. Hence,

$$
\begin{aligned}
I(W; U | \widetilde{Z}, V) &= \sum_{i=1}^{K} I(W; U^{i,V_i} | \widetilde{Z}, V, U^{[i-1]}) + I(U^{[i-1]}; U^{i,V_i} | \widetilde{Z}, V) \\
&= \sum_{i=1}^{K} I(W, U^{[i-1]}; U^{i,V_i} | \widetilde{Z}, V) \\
&= \sum_{i=1}^{K} I(W; U^{i,V_i} | \widetilde{Z}, V) + I(U^{[i-1]}; U^{i,V_i} | \widetilde{Z}, V, W) \\
&\geq \sum_{i=1}^{K} I(W; U^{i,V_i} | \widetilde{Z}, V). \tag{28}
\end{aligned}
$$

Thus, plugging Eq. (28) into Eq. (27), we have

$$
\sqrt{\frac{2}{Kn} \sum_{i=1}^{K} \sum_{j=1}^{n} I(W; U_j^{i,V_i} | \widetilde{Z}^i, V_i)} \leq \sqrt{\frac{2I(W; U | \widetilde{Z}, V)}{Kn}}. \tag{29}
$$

Combining Eq. (26) and Eq. (29) together will complete the proof. □

### D.5. Proof of Corollary 3.2

*Proof.* Since all data in $\widetilde{Z}$ are i.i.d., the conditional distribution $P_{W|V_i,\widetilde{Z}^i,U^i}$ is now invariant to $V_i$, namely $P_{W|V_i=0,\widetilde{Z}^i,U^i} = P_{W|V_i=1,\widetilde{Z}^i,U^i} = \frac{1}{2}P_{W|V_i=0,\widetilde{Z}^i,U^i} + \frac{1}{2}P_{W|V_i=1,\widetilde{Z}^i,U^i} = P_{W|\widetilde{Z}^i,U^i}$. Consequently,

$$I(W;V_i|\widetilde{Z}^i,U^i) = \mathbb{E}_{\widetilde{Z}^i,U^i}\left[\int_{w,v_i} dP(w,v_i|\widetilde{Z}^i,U^i)\log\frac{dP(w|v_i,\widetilde{Z}^i,U^i)}{dP(w|\widetilde{Z}^i,U^i)}\right] = 0,$$

which can imply the first term in Corollary 3.1 vanish. $\qquad\square$

### D.6. Proof of Theorem 3.2

*Proof.* Again, we start from bounding the participation gap. We let

$$g(\tilde{z},u,w,v) = \frac{K-1}{2}\left(\frac{1}{K}\sum_{i=1}^{K}(-1)^{v_i}\left(\ell(w,\tilde{z}^{i,1}_{1,\bar{u}^{i,1}_1}) - \ell(w,\tilde{z}^{i,0}_{1,\bar{u}^{i,0}_1})\right)\right)^2.$$

Then, we apply Lemma B.1

$$\mathbb{E}_{W|V,\widetilde{Z},U}\left[g(\widetilde{Z},U,W,V)\right] \le \mathrm{D}_{\mathrm{KL}}\left(P_{W|\widetilde{Z},U,V}||P_{W|\widetilde{Z},U}\right) + \log\mathbb{E}_{W|\widetilde{Z},U}\left[e^{g(\widetilde{Z},U,W,V)}\right]. \tag{30}$$

Let $V'$ be an independent copy of $V$. By Markov's inequality, we have the following bound with the probability at least $1-\delta$ under the draw of $(\widetilde{Z},U,V')$

$$\mathbb{E}_{W|V,\widetilde{Z},U}\left[g(\widetilde{Z},U,W,V)\right] \le \mathrm{D}_{\mathrm{KL}}\left(P_{W|\widetilde{Z},U,V}||P_{W|\widetilde{Z},U}\right) + \log\frac{\mathbb{E}_{W,\widetilde{Z},U}\mathbb{E}_{V'}\left[e^{g(\widetilde{Z},U,W,V')}\right]}{\delta}. \tag{31}$$

Since $V'$ is independent of $W, \widetilde{Z}$ and $U$, the random variable $\zeta = \frac{1}{K}\sum_{i=1}^{K}(-1)^{V'_i}\left(\ell(w,\tilde{z}^{i,1}_{1,\bar{u}^{i,1}_1}) - \gamma(w,\tilde{z}^{i,0}_{1,\bar{u}^{i,0}_1})\right)$ has zero mean for any given $(w,\tilde{z},u)$, namely $\mathbb{E}_{V'}[\zeta] = 0$. Additionally, for the fixed $(w,\tilde{z},u)$, $\zeta$ is the the arithmetic average of $K$ independent terms, each with bounded range $[-1,1]$. Hence, $\zeta$ is a sub-Gaussian random variable with variance proxy $1/\sqrt{K}$. By Wainwright (2019, Thm. 2.6.(IV)), we have

$$\mathbb{E}_{V'}\left[e^{\frac{K-1}{2}\zeta^2}\right] \le \sqrt{K}.$$

Plugging the above inequality into Eq. (31), we have

$$\mathbb{E}_{W|V,\widetilde{Z},U}\left[g(\widetilde{Z},U,W,V)\right] \le \mathrm{D}_{\mathrm{KL}}\left(P_{W|\widetilde{Z},U,V}||P_{W|\widetilde{Z},U}\right) + \log\frac{\sqrt{K}}{\delta}.$$

Then by Jensen's inequality,

$$\frac{K-1}{2}\left(\mathbb{E}_{W|V,\widetilde{Z},U}\left[\frac{1}{K}\sum_{i=1}^{K}(-1)^{V_i}\left(\ell(W,\widetilde{Z}^{i,1}_{1,\overline{U}^{i,1}_1}) - \ell(W,\widetilde{Z}^{i,0}_{1,\overline{U}^{i,0}_1})\right)\right]\right)^2 \le \mathrm{D}_{\mathrm{KL}}\left(P_{W|\widetilde{Z},U,V}||P_{W|\widetilde{Z},U}\right) + \log\frac{\sqrt{K}}{\delta}.$$

Consequently, with probability at least $1-\delta$,

$$\left|\mathbb{E}_{W|\widetilde{Z},U,V}\left[L_{\mathcal{D}}(W) - \frac{1}{K}\sum_{i=1}^{K}\ell(W,\widetilde{Z}^{i,V_i}_{1,\overline{U}^{i,V_i}_1})\right]\right| \le \sqrt{\frac{2\mathrm{D}_{\mathrm{KL}}\left(P_{W|\widetilde{Z},U,V}||P_{W|\widetilde{Z},U}\right) + 2\log\frac{\sqrt{K}}{\delta}}{K-1}}. \tag{32}$$

Furthermore, for out-of-sample gap, we let

$$h(\tilde{z},u,w,v) = \frac{nK-1}{2}\left(\frac{1}{Kn}\sum_{i=1}^{K}\sum_{j=1}^{n}\mathbb{E}_{\tilde{z}^i,v_i,u_j^{i,v_i},w}\left[(-1)^{u_j^{i,v_i}}\left(\ell(w,\tilde{z}^{i,v_i}_{j,1}) - \ell(w,\tilde{z}^{i,v_i}_{j,0})\right)\right]\right)^2.$$

Applying Lemma B.1

$$\mathbb{E}_{W|V,\widetilde{Z},U}\left[h(\widetilde{Z},U,W,V)\right] \leq D_{\mathrm{KL}}\left(P_{W|\widetilde{Z},U,V}||P_{W|\widetilde{Z},V}\right) + \log \mathbb{E}_{W|\widetilde{Z},V}\left[e^{h(\widetilde{Z},U,W,V)}\right]. \tag{33}$$

Let $U'$ be an independent copy of $U$. By Markov's inequality, we have the following with the probability at least $1 - \delta$ under the draw of $(\widetilde{Z}, U', V)$

$$\mathbb{E}_{W|V,\widetilde{Z},U}\left[h(\widetilde{Z},U,W,V)\right] \leq D_{\mathrm{KL}}\left(P_{W|\widetilde{Z},U,V}||P_{W|\widetilde{Z},V}\right) + \log \frac{\mathbb{E}_{W,\widetilde{Z},V}\mathbb{E}_{U'}\left[e^{h(\widetilde{Z},U',W,V)}\right]}{\delta}. \tag{34}$$

Since $U'$ is independent of $W, \widetilde{Z}$ and $V$, the random variable $\zeta' = \frac{1}{Kn}\sum_{i=1}^{K}\sum_{j=1}^{n}(-1)^{U_j^{i,v_i}}\left(\ell(w, \tilde{z}_{j,1}^{i,v_i}) - \ell(w, \tilde{z}_{j,0}^{i,v_i})\right)$ has zero mean for any given $(w, \tilde{z}, v)$, namely $\mathbb{E}_{U'}[\zeta'] = 0$. Additionally, for the fixed $(w, \tilde{z}, v)$, $\zeta'$ is the the arithmetic average of $Kn$ independent terms, each with bounded range $[-1, 1]$. Hence, $\zeta'$ is a sub-Gaussian random variable with variance proxy $1/\sqrt{Kn}$. By Wainwright (2019, Thm. 2.6.(IV)), we have

$$\mathbb{E}_{V'}\left[e^{\frac{Kn-1}{2}\zeta'^2}\right] \leq \sqrt{Kn}.$$

Plugging the above inequality into Eq. (34), we have

$$\mathbb{E}_{W|V,\widetilde{Z},U}\left[h(\widetilde{Z},U,W,V)\right] \leq D_{\mathrm{KL}}\left(P_{W|\widetilde{Z},U,V}||P_{W|\widetilde{Z},V}\right) + \log \frac{\sqrt{Kn}}{\delta}.$$

By Jensen's inequality,

$$\frac{Kn-1}{2}\left(\mathbb{E}_{W|V,\widetilde{Z},U}\left[\frac{1}{Kn}\sum_{i=1}^{K}\sum_{j=1}^{n}(-1)^{U_j^{i,V_i}}\left(\ell(W, \widetilde{Z}_{j,1}^{i,V_i}) - \ell(W, \widetilde{Z}_{j,0}^{i,V_i})\right)\right]\right)^2 \leq D_{\mathrm{KL}}\left(P_{W|\widetilde{Z},U,V}||P_{W|\widetilde{Z},V}\right) + \log \frac{\sqrt{Kn}}{\delta}.$$

Therefore, with probability at least $1 - \delta$,

$$\left|\mathbb{E}_{W|V,\widetilde{Z},U}\left[\frac{1}{Kn}\sum_{i=1}^{K}\sum_{j=1}^{n}(-1)^{U_j^{i,V_i}}\left(\ell(W, \widetilde{Z}_{j,1}^{i,V_i}) - \ell(W, \widetilde{Z}_{j,0}^{i,V_i})\right)\right]\right| \leq \sqrt{\frac{2D_{\mathrm{KL}}\left(P_{W|\widetilde{Z},U,V}||P_{W|\widetilde{Z},V}\right) + 2\log\frac{\sqrt{Kn}}{\delta}}{Kn-1}}. \tag{35}$$

Combining Eq. (32) and Eq. (35), with probability at least $1 - 2\delta$ under the draw of $(\widetilde{Z}, U, V)$, we have

$$\left|\mathbb{E}_{W|\widetilde{Z},U,V}\left[L_{\mathcal{D}}(W) - L_S(W)\right]\right|$$

$$\leq \left|\mathbb{E}_{W|\widetilde{Z},U,V}\left[L_{\mathcal{D}}(W) - \frac{1}{K}\sum_{i=1}^{K}\ell(W, \widetilde{Z}_{1,\overline{U}_1^{i,V_i}}^{i,V_i})\right]\right| + \left|\mathbb{E}_{W|\widetilde{Z},U,V}\left[\frac{1}{Kn}\sum_{i=1}^{K}\sum_{j=1}^{n}\left(\ell(W, \widetilde{Z}_{j,\overline{U}_j^{i,V_i}}^{i,V_i}) - \ell(W, \widetilde{Z}_{j,U_j^{i,V_i}}^{i,V_i})\right)\right]\right|$$

$$\leq \sqrt{\frac{2D_{\mathrm{KL}}\left(P_{W|\widetilde{Z},U,V}||P_{W|\widetilde{Z},U}\right) + 2\log\frac{\sqrt{K}}{\delta}}{K-1}} + \sqrt{\frac{2D_{\mathrm{KL}}\left(P_{W|\widetilde{Z},U,V}||P_{W|\widetilde{Z},V}\right) + 2\log\frac{\sqrt{Kn}}{\delta}}{Kn-1}}.$$

Finally, let $\delta \to \delta/2$ will complete the proof. $\qquad\square$

### D.7. Proof of Theorem 3.3

*Proof.* We first notice that

$$
\begin{aligned}
\mathcal{E}_{ER}(\mathcal{A}) =& \mathbb{E}_W\left[L_\mathcal{D}(W)\right] - \mathbb{E}_{\mu\sim\mathcal{D}}\mathbb{E}_{Z\sim\mu}\left[\ell(w^*, Z)\right]\\
=& \mathbb{E}_W\left[L_\mathcal{D}(W)\right] - \mathbb{E}_{W,S}\left[L_S(W)\right] + \mathbb{E}_{W,S}\left[L_S(W)\right] - \mathbb{E}_{\mu\sim\mathcal{D}}\mathbb{E}_{Z\sim\mu}\left[\ell(w^*, Z)\right]\\
\leq& \mathbb{E}_W\left[L_\mathcal{D}(W)\right] - \mathbb{E}_{W,S}\left[L_S(W)\right] + \mathbb{E}_S\left[L_S(w^*)\right] - \mathbb{E}_{\mu\sim\mathcal{D}}\mathbb{E}_{Z\sim\mu}\left[\ell(w^*, Z)\right]\\
=& \mathcal{E}_\mathcal{D}(\mathcal{A}),
\end{aligned}
$$

where the first inequality is due to the fact that $W$ is the empirical risk minimizer of $S$, and the last equality holds because $\mathbb{E}_S\left[L_S(w^*)\right] = \frac{1}{Kn}\sum_{i=1}^{K}\sum_{j=1}^{n}\mathbb{E}_{Z_{i,j}}\left[\ell(w^*, Z_{i,j})\right] = \mathbb{E}_Z\left[\ell(w^*, Z)\right]$.

Recall the generalization bound in Corollary 3.1,

$$
\mathcal{E}_{ER}(\mathcal{A}) \leq \mathcal{E}_\mathcal{D}(\mathcal{A}) \leq |\mathcal{E}_\mathcal{D}(\mathcal{A})| \leq \sqrt{\frac{2I(W;V|\widetilde{Z}, U)}{K}} + \sqrt{\frac{2I(W;U|\widetilde{Z}, V)}{Kn}}.
$$

This gives us the expected excess risk bound.

For the high-probability bound,

$$
\begin{aligned}
&\mathbb{E}_{W|\widetilde{Z},U,V}\left[L_\mathcal{D}(W) - L_S(W)\right]\\
=&\mathbb{E}_{W|\widetilde{Z},U,V}\left[L_\mathcal{D}(W) - \mathbb{E}_{\mu\sim\mathcal{D}}\mathbb{E}_{Z\sim\mu}\left[\ell(w^*, Z)\right] + \mathbb{E}_{\mu\sim\mathcal{D}}\mathbb{E}_{Z\sim\mu}\left[\ell(w^*, Z)\right] - L_S(W)\right]\\
=&\mathbb{E}_{W|\widetilde{Z},U,V}\left[L_\mathcal{D}(W) - \mathbb{E}_{\mu\sim\mathcal{D}}\mathbb{E}_{Z\sim\mu}\left[\ell(w^*, Z)\right]\right] - \underbrace{(\mathbb{E}_{W|\widetilde{Z},U,V}\left[L_S(W) - \mathbb{E}_{\mu\sim\mathcal{D}}\mathbb{E}_{Z\sim\mu}\left[\ell(w^*, Z)\right]\right])}_{B_1}.
\end{aligned}
$$

Hence,

$$
\mathbb{E}_{W|\widetilde{Z},U,V}\left[L_\mathcal{D}(W) - \mathbb{E}_{\mu\sim\mathcal{D}}\mathbb{E}_{Z\sim\mu}\left[\ell(w^*, Z)\right]\right] = \mathbb{E}_{W|\widetilde{Z},U,V}\left[L_\mathcal{D}(W) - L_S(W)\right] + B_1.
$$

The first gap in RHS, $\mathbb{E}_{W|\widetilde{Z},U,V}\left[L_\mathcal{D}(W) - L_S(W)\right]$, can be upper bounded by Theorem 3.2.

We further process the $B_1$ term,

$$
\begin{aligned}
B_1 =& \mathbb{E}_{W|\widetilde{Z},U,V}\left[L_S(W) - \mathbb{E}_S\left[L_S(w^*)\right] + \mathbb{E}_S\left[L_S(w^*)\right] - \mathbb{E}_{\mu\sim\mathcal{D}}\mathbb{E}_{Z\sim\mu}\left[\ell(w^*, Z)\right]\right]\\
\leq& \mathbb{E}_{W|\widetilde{Z},U,V}\left[L_S(w^*) - \mathbb{E}_{\mu\sim\mathcal{D}}\mathbb{E}_{Z\sim\mu}\left[\ell(w^*, Z)\right]\right]\\
=& L_S(w^*) - \mathbb{E}_{\mu\sim\mathcal{D}}\mathbb{E}_{Z\sim\mu}\left[\ell(w^*, Z)\right]\\
=& \frac{1}{nK}\sum_{i=1}^{n}\sum_{j=1}^{K}\ell(w^*, Z_{i,j}) - \mathbb{E}_Z\left[\ell(w^*, Z)\right].
\end{aligned}
$$

where the inequality holds because $W$ is the empirical risk minimizer of $S$.

Note that each $Z_{i,j}$ is independently drawn. Therefore, by Hoeffding's inequality (cf. Lemma B.3), we have, with probability at least $1 - \delta$,

$$
B_1 \leq \sqrt{\frac{\log\frac{2}{\delta}}{2nK}}.
$$

This completes the proof. $\qquad\square$

## E. Omitted Proof in Section 4

### E.1. Proof of Theorem 4.1

To prove Theorem 4.1, we first need the following lemma as the main ingredient.

**Lemma E.1.** *Let the constants $C_1, C_2 > 0$. The following equations hold for the weighted participation gap and weighted out-of-sample gap,*

$$\mathbb{E}_W\left[L_\mathcal{D}(W)\right] - (1+C_1)\mathbb{E}_{W,\mu_{[K]}}\left[L_{\mu_{[K]}}(W)\right] = \frac{2+C_1}{K}\sum_{i=1}^{K}\mathbb{E}_{\bar{L}_i^+,\tilde{\varepsilon}_i}\left[\tilde{\varepsilon}_i\bar{L}_i^+\right], \tag{36}$$

$$\mathbb{E}_{W,\mu_{[K]}}\left[L_{\mu_{[K]}}(W)\right] - (1+C_2)\mathbb{E}_{W,S}\left[L_S(W)\right] = \frac{2+C_2}{Kn}\sum_{i=1}^{K}\sum_{j=1}^{n}\mathbb{E}_{L_j^{+i},\tilde{\varepsilon}_j^{i,V_i},V_i}\left[\tilde{\varepsilon}_j^{i,V_i}L_j^{+i}\right], \tag{37}$$

*respectively, where $\tilde{\varepsilon}_i = (-1)^{\overline{V}_i} - \frac{C_1}{C_1+2}$ and $\tilde{\varepsilon}_j^{i,V_i} = (-1)^{\overline{U}_j^{i,V_i}} - \frac{C_1}{C_1+2}$ are two* shifted *Rademacher variable with the same mean $-\frac{C_1}{C_1+2}$.*

*Proof of Lemma E.1.* Let $\bar{L}_i^- = \ell(W, \widetilde{Z}_{1,\overline{U}^{i,1}_1}^{i,1})$, then

$$\mathbb{E}_W\left[L_\mathcal{D}(W)\right] - (1+C_1)\mathbb{E}_{W,\mu_{[K]}}\left[L_{\mu_{[K]}}(W)\right]$$

$$= \frac{1}{K}\sum_{i=1}^{K}\mathbb{E}_{\widetilde{Z}^i,U^i,W,V_i}\left[\ell(W,\widetilde{Z}_{1,\overline{U}_1^i,\overline{V}_i}^{i,\overline{V}_i}) - (1+C_1)\ell(W,\widetilde{Z}_{1,\overline{U}_1^i,V_i}^{i,V_i})\right]$$

$$= \frac{1}{K}\sum_{i=1}^{K}\mathbb{E}_{\widetilde{Z}^i,U^i,W,V_i}\left[\left(1+\frac{C_1}{2}\right)\left(\ell(W,\widetilde{Z}_{1,\overline{U}_1^i,\overline{V}_i}^{i,\overline{V}_i}) - \ell(W,\widetilde{Z}_{1,\overline{U}_1^i,V_i}^{i,V_i})\right) - \frac{C_1}{2}\ell(W,\widetilde{Z}_{1,\overline{U}_1^i,\overline{V}_i}^{i,\overline{V}_i}) - \frac{C_1}{2}\ell(W,\widetilde{Z}_{1,\overline{U}_1^i,V_i}^{i,V_i})\right]$$

$$= \frac{2+C_1}{2K}\sum_{i=1}^{K}\left[\mathbb{E}_{\bar{L}_i^-,V_i}\left[(-1)^{V_i}\bar{L}_i^- - \frac{C_1}{C_1+2}\bar{L}_i^-\right] + \mathbb{E}_{\bar{L}_i^+,V_i}\left[-(-1)^{V_i}\bar{L}_i^+ - \frac{C_1}{C_1+2}\bar{L}_i^+\right]\right].$$

By the symmetric property of superclient, it is easy to see that, no matter in the homogeneous setting or the heterogeneous setting, $\mathbb{E}_{\bar{L}_i^-,V_i}\left[(-1)^{V_i}\bar{L}_i^- - \frac{C_1}{C_1+2}\bar{L}_i^-\right] = \mathbb{E}_{\bar{L}_i^+,V_i}\left[-(-1)^{V_i}\bar{L}_i^+ - \frac{C_1}{C_1+2}\bar{L}_i^+\right]$ holds, we then have

$$\mathbb{E}_W\left[L_\mathcal{D}(W)\right] - (1+C_1)\mathbb{E}_{W,\mu_{[K]}}\left[L_{\mu_{[K]}}(W)\right]$$

$$= \frac{2+C_1}{K}\sum_{i=1}^{K}\mathbb{E}_{\bar{L}_i^+,\overline{V}_i}\left[(-1)^{\overline{V}_i}\bar{L}_i^+ - \frac{C_1}{C_1+2}\bar{L}_i^+\right]$$

$$= \frac{2+C_1}{K}\sum_{i=1}^{K}\mathbb{E}_{\bar{L}_i^+,\tilde{\varepsilon}_i}\left[\tilde{\varepsilon}_i\bar{L}_i^+\right]. \tag{38}$$

For the second part, the decomposition is nearly the same. In particular, let $L_j^{i-} = \ell(W,\widetilde{Z}_{j,1}^{i,V_i})$, then

$$\mathbb{E}_{W,\mu_{[K]}}\left[L_{\mu_{[K]}}(W)\right] - (1+C_2)\mathbb{E}_{W,S}\left[L_S(W)\right]$$

$$= \frac{1}{Kn}\sum_{i=1}^{K}\sum_{j=1}^{n}\mathbb{E}_{\widetilde{Z}^i,V_i,U_j^{i,V_i},W}\left[\ell(W,\widetilde{Z}_{j,\overline{U}_j^{i,V_i}}^{i,V_i}) - \ell(W,\widetilde{Z}_{j,U_j^{i,V_i}}^{i,V_i})\right]$$

$$= \frac{2+C_2}{2Kn}\sum_{i=1}^{K}\sum_{j=1}^{n}\left[\mathbb{E}_{L_j^{i-},V_i,U_j^{i,V_i}}\left[(-1)^{U_j^{i,V_i}}L_j^{i-} - \frac{C_2}{C_2+2}L_j^{i-}\right] + \mathbb{E}_{L_j^{i+},V_i,U_j^{i,V_i}}\left[-(-1)^{U_j^{i,V_i}}L_j^{i+} - \frac{C_2}{C_2+2}L_j^{i+}\right]\right]$$

$$= \frac{2+C_2}{Kn}\sum_{i=1}^{K}\sum_{j=1}^{n}\mathbb{E}_{L_j^{i+},V_i,U_j^{i,V_i}}\left[(-1)^{\overline{U}_j^{i,V_i}}L_j^{i+} - \frac{C_2}{C_2+2}L_j^{i+}\right]$$

$$= \frac{2+C_2}{Kn}\sum_{i=1}^{K}\sum_{j=1}^{n}\mathbb{E}_{L_j^{i+},V_i,\tilde{\varepsilon}_j^{i,V_i}}\left[\tilde{\varepsilon}_j^{i,V_i}L_j^{i+}\right].$$

This completes the proof. □

We are now ready to prove Theorem 4.1.

*Proof of Theorem 4.1.* Recall Lemma B.1, let $g = t(C_1 + 2)\tilde{\varepsilon}_i \bar{L}_i^+$, and let $\tilde{\varepsilon}_i'$ be an independent copy of $\tilde{\varepsilon}_i$, then

$$
\begin{aligned}
I(\bar{L}_i^+; V_i) = I(\bar{L}_i^+; \tilde{\varepsilon}_i) = & D_{\text{KL}}\left(P_{\bar{L}_i^+, \tilde{\varepsilon}_i} \| P_{\bar{L}_i^+} P_{\tilde{\varepsilon}_i'}\right) \\
\geq & \sup_{t>0} \mathbb{E}_{\bar{L}_i^+, \tilde{\varepsilon}_i}\left[t(C_1+2)\tilde{\varepsilon}_i \bar{L}_i^+\right] - \log \mathbb{E}_{\bar{L}_i^+, \tilde{\varepsilon}_i'}\left[e^{t(C_1+2)\tilde{\varepsilon}_i' \bar{L}_i^+}\right] \\
= & \sup_{t>0} \mathbb{E}_{\bar{L}_i^+, \tilde{\varepsilon}_i}\left[t(C_1+2)\tilde{\varepsilon}_i \bar{L}_i^+\right] - \log \frac{\mathbb{E}_{\bar{L}_i^+}\left[e^{-2t(C_1+1)\bar{L}_i^+} + e^{2t\bar{L}_i^+}\right]}{2},
\end{aligned}
\tag{39}
$$

where we use the fact that $\tilde{\varepsilon}_i'$ is independent of $\bar{L}_i^+$, and $P(\tilde{\varepsilon}_i = \frac{2}{C_1+2}) = P(\tilde{\varepsilon}_i = \frac{-2(C_1+1)}{C_1+2}) = \frac{1}{2}$.

Notice that $e^{-2t(C_1+1)\bar{L}_i^+} + e^{2t\bar{L}_i^+}$ is the summation of two convex function, which is still a convex function, so the maximum value of this function is achieved at the endpoints of the bounded domain. Recall that $\bar{L}_i^+ \in [0, 1]$, we now consider two cases:

i) when $\bar{L}_i^+ = 0$, we have $e^{-2t(C_1+1)\bar{L}_i^+} + e^{2t\bar{L}_i^+} = 2$;

ii) when $\bar{L}_i^+ = 1$, we select $t$ such that $e^{-2t(C_1+1)} + e^{2t} \leq 2$. Note that this inequality implies that $t \leq \frac{\log 2}{2}$.

Hence, let $t = C_3$, and let the values of $C_1, C_3$ be taken from the domain of $\{C_1, C_3 | C_1, C_3 > 0, e^{-2C_3(C_1+1)} + e^{2C_3} \leq 2\}$, so the inequality

$$
\frac{\mathbb{E}_{\bar{L}_i^+}\left[e^{-2C_3(C_1+1)\bar{L}_i^+} + e^{2C_3\bar{L}_i^+}\right]}{2} \leq 1
\tag{40}
$$

will hold. Under this condition, by re-arranging the terms in Eq. (39), we have

$$
(C_1 + 2)\mathbb{E}_{\bar{L}_i^+, \tilde{\varepsilon}_i}\left[\tilde{\varepsilon}_i \bar{L}_i^+\right] \leq \frac{I(\bar{L}_i^+; V_i)}{C_3}.
$$

Plugging the inequality above into Eq. (37), we have

$$
\mathbb{E}_W\left[L_{\mathcal{D}}(W)\right] - (1 + C_1)\mathbb{E}_{W, \mu_{[K]}}\left[L_{\mu_{[K]}}(W)\right] = \frac{2 + C_1}{K} \sum_{i=1}^{K} \mathbb{E}_{\bar{L}_i^+, \tilde{\varepsilon}_i}\left[\tilde{\varepsilon}_i \bar{L}_i^+\right] \leq \sum_{i=1}^{K} \frac{I(\bar{L}_i^+; V_i)}{C_3 m}.
$$

Thus, the following inequality can be obtained,

$$
\mathbb{E}_W\left[L_{\mathcal{D}}(W)\right] \leq \min_{C_1, C_3 > 0, e^{2C_3} + e^{-2C_3(C_1+1)} \leq 2} (1 + C_1)\mathbb{E}_{W, \mu_{[K]}}\left[L_{\mu_{[K]}}(W)\right] + \sum_{i=1}^{K} \frac{I(\bar{L}_i^+; V_i)}{C_3 m}.
\tag{41}
$$

Following the similar development, we can also obtain

$$
\begin{aligned}
& \mathbb{E}_{W, \mu_{[K]}}\left[L_{\mu_{[K]}}(W)\right] - (1 + C_2)\mathbb{E}_{W, S}\left[L_S(W)\right] \\
= & \frac{2 + C_2}{Kn} \sum_{i=1}^{K} \sum_{j=1}^{n} \mathbb{E}_{L_j^{+i}, \tilde{\varepsilon}_j^{i, V_i}, V_i}\left[\tilde{\varepsilon}_j^{i, V_i} L_j^{+i}\right] \\
\leq & \sum_{i=1}^{K} \sum_{j=1}^{n} \frac{I(L_j^{+i}; U_j^{i, V_i})}{C_4 Kn}.
\end{aligned}
$$

This is equivalent to

$$
\mathbb{E}_{W, \mu_{[K]}}\left[L_{\mu_{[K]}}(W)\right] \leq \min_{C_2, C_4 > 0, e^{2C_4} + e^{-2C_4(C_2+1)} \leq 2} (1 + C_2)\mathbb{E}_{W, S}\left[L_S(W)\right] + \sum_{i=1}^{K} \sum_{j=1}^{n} \frac{I(L_j^{+i}; U_j^{i, V_i})}{C_4 Kn}.
\tag{42}
$$

Finally, substituting Eq. (42) into Eq. (41) and re-assigning $C_1 = C_1 + 1$ and $C_2 = C_2 + 1$ will complete the proof. □

**E.2. Proof of Corollary 4.1**

*Proof.* In fact, the generalization bound in Corollary 4.1 can be similarly proved as in Theorem 4.1 with unpacking the random variables $\bar{L}_i^+ = \ell(W, \widetilde{Z}_{1,\overline{U}_1^{i,0}}^{i,0})$ and $L_j^{i+} = \ell(W, \widetilde{Z}_{j,0}^{i,V_i})$ during the development, which are functions of $(W, \widetilde{Z}^i, U^i)$ and $(W, \widetilde{Z}^i, V_i)$, respectively.

Alternatively, notice that

$$I(\bar{L}_i^+; V_i | \widetilde{Z}^i, U^i) + I(V_i; \widetilde{Z}^i, U^i) = I(\bar{L}_i^+, \widetilde{Z}^i, U^i; V_i) = I(\bar{L}_i^+; V_i) + I(\widetilde{Z}^i, U^i; V_i | \bar{L}_i^+).$$

Since $I(V_i; \widetilde{Z}^i, U^i) = 0$, we have $I(\bar{L}_i^+; V_i) \leq I(\bar{L}_i^+; V_i | \widetilde{Z}^i, U^i)$. By the data-processing inequality, we further obtain $I(\bar{L}_i^+; V_i | \widetilde{Z}^i, U^i) \leq I(W; V_i | \widetilde{Z}^i, U^i)$.

Analogously, we can also obtain $I(L_j^{i+}; U_j^{i,V_i} | V_i) \leq I(W; U_j^{i,V_i} | V_i, \widetilde{Z}^i)$. The remaining steps are the same as in the proof of Corollary 3.1. $\square$

# F. Omitted Proof in Section 5

## F.1. Proof of Theorem 5.1

The most important ingredient for proving Theorem 5.1 is the following lemma.

**Lemma F.1.** *Let $\ell(w, z) = D_f(w, z)$, we have*

$$\mathcal{E}_{PG}(\mathcal{A}) = \frac{1}{K^2} \sum_{i=1}^{K} \mathbb{E}_{\widetilde{Z}^i, U^i, W_i, V_i} \left[ (-1)^{V_i} \left( \ell(W_i, \widetilde{Z}_{1,\overline{U}_1^{i,1}}^{i,1}) - \ell(W_i, \widetilde{Z}_{1,\overline{U}_1^{i,0}}^{i,0}) \right) \right]$$

$$\mathcal{E}_{OG}(\mathcal{A}) = \frac{1}{K^2 n} \sum_{i=1}^{K} \sum_{j=1}^{n} \mathbb{E}_{\widetilde{Z}^i, V_i, U_j^{i,V_i}, W_i} \left[ (-1)^{U_j^{i,V_i}} \left( \ell(W_i, \widetilde{Z}_{j,1}^{i,V_i}) - \ell(W_i, \widetilde{Z}_{j,0}^{i,V_i}) \right) \right].$$

*Proof of Lemma F.1.* Considering the participation gap. The $i$-th participating client $\mu_i$ uses a training sample $S_i$ to train a local model $W_i$, then if this client is replaced by another client, $\mu_i'$, which uses its own training sample $S_i'$, then the provided local model is denoted as $W^{(i)}$. In this case, we denote the new aggregation model $W^{(i)} = \frac{1}{K} \left( \sum_{k \neq i} W_k + W_i' \right)$.

Then, let $\bar{Z}_i' = \widetilde{Z}_{1,\overline{U}_1^{i,\overline{V}_i}}^{i,\overline{V}_i}$ be the testing data from non-participating client $\tilde{\mu}_{i,\overline{V}_i}$ and let $\bar{Z}_i = \widetilde{Z}_{1,\overline{U}_1^{i,V_i}}^{i,V_i}$ be the testing data from participating client $\tilde{\mu}_{i,V_i}$. Notice that

$$\ell(W, \bar{Z}_i') - \ell(W, \bar{Z}_i) = \ell(\mathcal{A}(S), \bar{Z}_i') - \ell(\mathcal{A}(S), \bar{Z}_i)$$
$$\ell(W, \bar{Z}_i') - \ell(W^{(i)}, \bar{Z}_i') = \ell(\mathcal{A}(S), \bar{Z}_i') - \ell(\mathcal{A}(S^i), \bar{Z}_i'),$$

where $S^i = (S \setminus S_i) \cup S_i'$.

Under our assumption that the $i$-th client always uses the same algorithm $\mathcal{A}_i$, the key observation is that $\mathbb{E}_{\mu_{[K]}, \bar{Z}_i, \mathcal{A}} \left[ \ell(\mathcal{A}(S), \bar{Z}_i) \right] = \mathbb{E}_{\mu_{[K] \setminus i}, \mu_i', \bar{Z}_i', \mathcal{A}} \left[ \ell(\mathcal{A}(S^i), \bar{Z}_i') \right]$, where, with an abuse of the notation, we also use $\mathcal{A}$ to denote the inherent randomness of algorithm. Then, we have the following

$$\mathcal{E}_{PG}(\mathcal{A}) = \frac{1}{K} \sum_{i=1}^{K} \mathbb{E}_{\widetilde{Z}^i, U^i, W, V_i} \left[ \ell(W, \bar{Z}_i') - \ell(W, \bar{Z}_i) \right]$$

$$= \frac{1}{K} \sum_{i=1}^{K} \mathbb{E}_{\widetilde{Z}^i, U^i, W, W^{(i)}, V_i} \left[ \ell(W, \bar{Z}_i') - \ell(W^{(i)}, \bar{Z}_i') \right].$$

Recall that $\ell(w, z) = \mathcal{D}_f(w, z)$, we have

$$\mathbb{E}\left[\ell(W, \bar{Z}'_i) - \ell(W^{(i)}, \bar{Z}'_i)\right]$$
$$=\mathbb{E}\left[f(W) - f(\bar{Z}'_i) - \langle \nabla f(\bar{Z}'_i), W - \bar{Z}'_i \rangle - \left(f(W^{(i)}) - f(\bar{Z}'_i) - \langle \nabla f(\bar{Z}'_i), W^{(i)} - \bar{Z}'_i \rangle\right)\right]$$
$$=\mathbb{E}\left[\langle \nabla f(\bar{Z}'_i), W^{(i)} - W \rangle\right],$$

where the last equation is due to the fact that $W$ and $W^{(i)}$ have the same marginal distribution, namely $\mathbb{E}_{\mathcal{D}, P_\mathcal{A}}\left[f(W)\right] = \mathbb{E}_{\mathcal{D}, P_\mathcal{A}}\left[f(W^{(i)})\right]$.

Since $W^{(i)} = \frac{1}{K}\left(\sum_{k \neq i} W_k + W'_i\right)$ and $W = \frac{1}{K}\sum_{i=1}^K W_i$, they only differ at $i$-th local model, we have $W^{(i)} - W = \frac{1}{K}\left(W'_i - W_i\right)$. Consequently,

$$\mathbb{E}\left[\ell(W, \bar{Z}'_i) - \ell(W^{(i)}, \bar{Z}'_i)\right] = \frac{1}{K}\mathbb{E}\left[\langle \nabla f(\bar{Z}'_i), W'_i - W_i \rangle\right]$$
$$= \frac{1}{K}\mathbb{E}\left[\langle \nabla f(\bar{Z}'_i), W'_i - \bar{Z}'_i - (W_i - \bar{Z}'_i) \rangle + f(\bar{Z}'_i) - f(\bar{Z}'_i) + f(W_i) - f(W'_i)\right]$$
$$= \frac{1}{K}\mathbb{E}\left[\ell(W_i, \bar{Z}'_i) - \ell(W_i, \bar{Z}_i)\right],$$

where we use $\mathbb{E}\left[f(W_i)\right] = \mathbb{E}\left[f(W'_i)\right]$ in the second equality.

Therefore,

$$\mathcal{E}_{PG}(\mathcal{A}) = \frac{1}{K^2}\sum_{i=1}^K \mathbb{E}_{\widetilde{Z}^i, U^i, W_i, V_i}\left[\left(\ell(W_i, \widetilde{Z}^{i, \overline{V}_i}_{1, \overline{U}^{i, \overline{V}_i}_1}) - \ell(W_i, \widetilde{Z}^{i, V_i}_{1, \overline{U}^{i, V_i}_1})\right)\right]$$
$$= \frac{1}{K^2}\sum_{i=1}^K \mathbb{E}_{\widetilde{Z}^i, U^i, W_i, V_i}\left[(-1)^{V_i}\left(\ell(W_i, \widetilde{Z}^{i,1}_{1, \overline{U}^{i,1}_1}) - \ell(W_i, \widetilde{Z}^{i,0}_{1, \overline{U}^{i,0}_1})\right)\right].$$

For out-of-sample gap, we now consider keeping the $i$-th participating client $\mu_i$ unchanged, but the $j$-th data in its training sample $S_i$ is replaced by another i.i.d. sampled data $Z'_{i,j} \sim \mu_i$, then the provided local model is denoted as $W'_{i,j}$. In this case, we denote the new aggregation model $W^{(i,j)} = \frac{1}{K}\left(\sum_{k \neq i} W_k + W'_{i,j}\right)$.

Recall that $\ell(w, z) = \mathcal{D}_f(w, z)$, for each participating client $\tilde{\mu}_{i, V_i}$, we have

$$\frac{1}{n}\sum_{j=1}^n \mathbb{E}\left[\ell(W, Z'_{i,j}) - \ell(W, Z_{i,j})\right]$$
$$= \frac{1}{n}\sum_{j=1}^n \mathbb{E}\left[\ell(W, Z'_{i,j}) - \ell(W^{(i,j)}, Z'_{i,j})\right]$$
$$= \frac{1}{n}\sum_{j=1}^n \mathbb{E}\left[f(W) - f(Z'_{i,j}) - \langle \nabla f(Z'_{i,j}), W - Z'_{i,j} \rangle - \left(f(W^{(i,j)}) - f(Z'_{i,j}) - \langle \nabla f(Z'_{i,j}), W^{(i,j)} - Z'_{i,j} \rangle\right)\right]$$
$$= \frac{1}{n}\sum_{j=1}^n \mathbb{E}\left[\langle \nabla f(Z'_{i,j}), W^{(i,j)} - W \rangle\right],$$

where the last equation is by $\mathbb{E}\left[f(W)\right] = \mathbb{E}\left[f(W^{(i,j)})\right]$.

Since $W^{(i,j)} = \frac{1}{K}\left(\sum_{k \neq i} W_k + W'_{i,j}\right)$ and $W = \frac{1}{K}\sum_{i=1}^K W_i$, they still only differ at $i$-th local model, we have $W^{(i,j)} - $

$W = \frac{1}{K}\left(W'_{i,j} - W_i\right)$. Consequently,

$$\frac{1}{n}\sum_{j=1}^{n}\mathbb{E}\left[\ell(W, Z'_{i,j}) - \ell(W, Z_{i,j})\right]$$

$$= \frac{1}{Kn}\sum_{j=1}^{n}\mathbb{E}\left[\langle\nabla f(Z'_{i,j}), W'_{i,j} - W_i\rangle\right]$$

$$= \frac{1}{Kn}\sum_{j=1}^{n}\mathbb{E}\left[\langle\nabla f(Z'_{i,j}), W'_{i,j} - Z'_{i,j} - (W_i - Z'_{i,j})\rangle + f(Z'_{i,j}) - f(Z'_{i,j}) + f(W_i) - f(W'_{i,j})\right]$$

$$= \frac{1}{nK}\sum_{j=1}^{n}\mathbb{E}\left[\ell(W_i, Z'_{i,j}) - \ell(W'_{i,j}, Z'_{i,j})\right]$$

$$= \frac{1}{nK}\sum_{j=1}^{n}\mathbb{E}\left[\ell(W_i, Z'_{i,j}) - \ell(W_i, Z_{i,j})\right],$$

where in the last equality we use the fact that $\mathbb{E}\left[\ell(W'_{i,j}, Z'_{i,j})\right] = \mathbb{E}\left[\ell(W_i, Z_{i,j})\right]$.

Therefore,

$$\mathcal{E}_{OG}(\mathcal{A}) = \frac{1}{Kn}\sum_{i=1}^{K}\sum_{j=1}^{n}\mathbb{E}_{W,\mu_i}\left[\ell(W, Z'_{i,j}) - \ell(W, Z_{i,j})\right]$$

$$= \frac{1}{K^2 n}\sum_{i=1}^{K}\sum_{j=1}^{n}\mathbb{E}_{\widetilde{Z}^i, V_i, U_j^{i,V_i}, W_i}\left[\ell(W_i, \widetilde{Z}_{j,\overline{U}_j^{i,V_i}}^{i,V_i}) - \ell(W_i, \widetilde{Z}_{j,U_j^{i,V_i}}^{i,V_i})\right]$$

$$= \frac{1}{K^2 n}\sum_{i=1}^{K}\sum_{j=1}^{n}\mathbb{E}_{\widetilde{Z}^i, V_i, U_j^{i,V_i}, W_i}\left[(-1)^{U_j^{i,V_i}}\left(\ell(W_i, \widetilde{Z}_{j,1}^{i,V_i}) - \ell(W_i, \widetilde{Z}_{j,0}^{i,V_i})\right)\right].$$

This completes the proof. □

*Proof of Theorem 5.1.* Having Lemma F.1 in hand, the development for proving the bound in Theorem 5.1 nearly follows the same procedure in the proof of Theorem 3.1.

The main difference lies in bounding the cumulant generating function, where Theorem 3.1 uses Lemma B.2 for bounded random vairable, i.e. Eq. (9) and Eq. (13), here we use the definition of sub-Gaussian random variable. We denote

$$A_1 = (-1)^{V'_i}\left(\ell(W_i, \tilde{z}_{1,\bar{u}_1^{i,1}}^{i,1}) - \ell(W_i, \tilde{z}_{1,\bar{u}_1^{i,0}}^{i,0})\right),$$

$$A_2 = (-1)^{U'_j^{i,v_i}}\left(\ell(W_i, \tilde{z}_{j,1}^{i,v_i}) - \ell(W_i, \tilde{z}_{j,0}^{i,v_i})\right).$$

Notice that due to the conditional independence between $W_i$ and $V'_i$, and the conditional independence $W_i$ and $U'_j^{i,v_i}$, $A_1$ and $A_2$ both have zero mean. Therefore, by the definition of sub-Gaussian random variable,

$$\mathbb{E}_{W_i, V'_i | \tilde{z}^i, u^i}\left[e^{tA_1}\right] \leq e^{\frac{t^2\sigma_i^2}{2}}, \tag{43}$$

$$\mathbb{E}_{W_i, U'_j^{i,v_i} | \tilde{z}^i, v_i}\left[e^{tA_2}\right] \leq e^{\frac{t^2\sigma_{i,j}^2}{2}}. \tag{44}$$

Eq. (43-44) are used to replace Eq. (9) and Eq. (13), respectively. The remaining steps are the same with the proof of Theorem 3.1, see Appendix D.2, that is, we can obtain that

$$\left|\mathbb{E}_{\widetilde{Z}^i, U^i, W_i, V_i}\left[(-1)^{V_i}\left(\ell(W_i, \widetilde{Z}_{1,\overline{U}_1^i}^{i,1}) - \ell(W_i, \widetilde{Z}_{1,\overline{U}_1^i}^{i,0})\right)\right]\right| \leq \mathbb{E}_{\widetilde{Z}^i, U^i}\sqrt{2I^{\widetilde{Z}^i, U^i}(W_i, V_i)},$$

$$\left|\mathbb{E}_{\widetilde{Z}^i, V_i, U_j^{i,V_i}, W_i}\left[(-1)^{U_j^{i,V_i}}\left(\ell(W_i, \widetilde{Z}_{j,1}^{i,V_i}) - \ell(W_i, \widetilde{Z}_{j,0}^{i,V_i})\right)\right]\right| \leq \mathbb{E}_{\widetilde{Z}^i, V_i}\sqrt{2I^{\widetilde{Z}^i, V_i}(W_i, U_j^{i,V_i})}.$$

Finally, plugging the above bounds into the inequalities in Lemma F.1 will complete the proof. □

### F.2. Proof of Theorem 5.2

*Proof.* First, the smoothness and strong convexity of loss function indicate that, for any $z$,

$$\text{Smoothness:} \quad \ell(w_1, z) \leq \ell(w_2, z) + \langle \nabla \ell(w_2, z), w_1 - w_2 \rangle + \frac{L}{2} ||w_1 - w_2||^2, \qquad \forall w_1, w_2 \in \mathcal{W}.$$

$$\text{Strong Convexity:} \quad \ell(w_1, z) \geq \ell(w_2, z) + \langle \nabla \ell(w_2, z), w_1 - w_2 \rangle + \frac{\alpha}{2} ||w_1 - w_2||^2, \qquad \forall w_1, w_2 \in \mathcal{W}.$$

The following development is inspired by Gholami & Seferoglu (2024).

For each participating client $\tilde{\mu}_{i, V_i}$, we have

$$
\begin{aligned}
\frac{1}{n} \sum_{j=1}^{n} \mathbb{E}\left[\ell(W, Z'_{i,j}) - \ell(W, Z_{i,j})\right] =& \frac{1}{n} \sum_{j=1}^{n} \mathbb{E}\left[\ell(W, Z'_{i,j}) - \ell(W^{(i,j)}, Z'_{i,j})\right] \\
\leq& \frac{1}{n} \sum_{j=1}^{n} \mathbb{E}\left[\langle \nabla \ell(W^{(i,j)}, Z'_{i,j}), W - W^{(i,j)} \rangle + \frac{L}{2} ||W - W^{(i,j)}||^2\right] \\
=& \frac{1}{Kn} \sum_{j=1}^{n} \mathbb{E}\left[\langle \nabla \ell(W^{(i,j)}, Z'_{i,j}), W_i - W'_{i,j} \rangle\right] + \frac{L}{2K^2 n} \sum_{j=1}^{n} \mathbb{E}\left[||W_i - W'_{i,j}||^2\right] \\
\leq& \frac{1}{Kn} \sum_{j=1}^{n} \sqrt{\mathbb{E}\left[||\nabla \ell(W^{(i,j)}, Z'_{i,j})||^2\right] \mathbb{E}\left[||W_i - W'_{i,j}||^2\right]} + \frac{L}{2K^2 n} \sum_{j=1}^{n} \mathbb{E}\left[||W_i - W'_{i,j}||^2\right],
\end{aligned}
\tag{45}
$$

where the first inequality is by the smoothness of $\ell(w, z)$ and the last inequality is by Cauchy-Schwarz inequality.

Again, by the smoothness property and $\nabla \ell(W'_{i,j}, Z'_{i,j}) = 0$, we have

$$
\begin{aligned}
\mathbb{E}\left[||\nabla \ell(W^{(i,j)}, Z'_{i,j})||^2\right] =& \mathbb{E}\left[||\nabla \ell(W^{(i,j)}, Z'_{i,j}) - \nabla \ell(W'_{i,j}, Z'_{i,j})||^2\right] \\
\leq& 2L \mathbb{E}\left[\ell(W^{(i,j)}, Z'_{i,j}) - \ell(W'_{i,j}, Z'_{i,j})\right] \\
=& 2L \mathbb{E}\left[\ell(W, Z_{i,j}) - \ell(W_i, Z_{i,j})\right].
\end{aligned}
\tag{46}
$$

In addition, by the strong convexity and $\nabla \ell(W'_{i,j}, Z'_{i,j}) = 0$, we have

$$
\begin{aligned}
\mathbb{E}\left[||W_i - W'_{i,j}||^2\right] \leq& \frac{2}{\alpha} \mathbb{E}\left[\ell(W_i, Z'_{i,j}) - \ell(W'_{i,j}, Z'_{i,j})\right] \\
=& \frac{2}{\alpha} \mathbb{E}\left[\ell(W_i, Z'_{i,j}) - \ell(W_i, Z_{i,j})\right].
\end{aligned}
\tag{47}
$$

Plugging Eq. (46-47) into Eq. (45), we have

$$
\begin{aligned}
& \frac{1}{n} \sum_{j=1}^{n} \mathbb{E}\left[\ell(W, Z'_{i,j}) - \ell(W, Z_{i,j})\right] \\
\leq& \frac{2}{Kn} \sum_{j=1}^{n} \sqrt{\frac{L}{\alpha} \mathbb{E}\left[\ell(W, Z_{i,j}) - \ell(W_i, Z_{i,j})\right] \mathbb{E}\left[\ell(W_i, Z'_{i,j}) - \ell(W_i, Z_{i,j})\right]} + \frac{L}{\alpha K^2 n} \sum_{j=1}^{n} \mathbb{E}\left[\ell(W_i, Z'_{i,j}) - \ell(W_i, Z_{i,j})\right] \\
\leq& \frac{2}{Kn} \sum_{j=1}^{n} \sqrt{\frac{L}{\alpha} \mathbb{E}\left[\ell(W, Z_{i,j}) - \ell(W_i, Z_{i,j})\right] \mathbb{E}\left[(-1)^{U_j^{i, V_i}} \left(\ell(W_i, \widetilde{Z}_{j,1}^{i, V_i}) - \ell(W_i, \widetilde{Z}_{j,0}^{i, V_i})\right)\right]} \\
& + \frac{L}{\alpha K^2 n} \sum_{j=1}^{n} \mathbb{E}\left[(-1)^{U_j^{i, V_i}} \left(\ell(W_i, \widetilde{Z}_{j,1}^{i, V_i}) - \ell(W_i, \widetilde{Z}_{j,0}^{i, V_i})\right)\right].
\end{aligned}
\tag{48}
$$

The remaining task is to bound $\mathbb{E}\left[(-1)^{U_j^{i,V_i}}\left(\ell(W_i,\widetilde{Z}_{j,1}^{i,V_i}) - \ell(W_i,\widetilde{Z}_{j,0}^{i,V_i})\right)\right]$. Recall that each $\mathcal{A}_i$ is an interpolating algorithm, we adapt similar techniques from Theorem 4.1, obtaining the following bound would be straightforward

$$\mathbb{E}\left[(-1)^{U_j^{i,V_i}}\left(\ell(W_i,\widetilde{Z}_{j,1}^{i,V_i}) - \ell(W_i,\widetilde{Z}_{j,0}^{i,V_i})\right)\right] \leq \frac{2I(W_i;U_j^{i,V_i}|\widetilde{Z}^i,V_i)}{\log 2},$$

Here, the constant $C_4 = \log 2/2$ arises from the fact that the empirical risk is zero (due to interpolation), allowing the multiplier $C_2$ in Eq. (42) to become arbitrarily large and thus broadening the optimization range for $C_4$. By solving the inequality $e^{-2C_4 C_2} + e^{2C_4} \leq 2$ in the limit $C_2 \to \infty$, we have $C_4 = \frac{\log 2}{2}$.

Consequently, we have

$$\begin{aligned}
\mathcal{E}_{OG}(\mathcal{A}) =& \frac{1}{K}\sum_{i=1}^{K}\left(\frac{1}{n}\sum_{j=1}^{n}\mathbb{E}\left[\ell(W,Z_{i,j}') - \ell(W,Z_{i,j})\right]\right) \\
\leq& \frac{2}{K^2 n}\sum_{i=1}^{K}\sum_{j=1}^{n}\sqrt{\frac{2L}{\alpha\log 2}\mathbb{E}\left[\ell(W,Z_{i,j})\right]I(W_i;U_j^{i,V_i}|\widetilde{Z}^i,V_i)} + \frac{2L}{\alpha K^3 n\log 2}\sum_{i=1}^{K}\sum_{j=1}^{n}I(W_i;U_j^{i,V_i}|\widetilde{Z}^i,V_i).
\end{aligned}$$

This completes the proof. $\square$

## G. Additional Result: Loss-Difference CMI Version of Theorem 3.1

We apply the loss-difference (LD)-CMI technique from Wang & Mao (2023b) to derive a tighter version of Theorem 3.1. Let $\bar{L}_i^{+} = \ell(W,\widetilde{Z}_{1,\overline{U}_1^{i,0}}^{i,0})$ and $\bar{L}_i^{-} = \ell(W,\widetilde{Z}_{1,\overline{U}_1^{i,1}}^{i,1})$. Let $L_j^{i+} = \ell(W,\widetilde{Z}_{j,0}^{i,V_i})$ and $L_j^{i-} = \ell(W,\widetilde{Z}_{j,1}^{i,V_i})$. Denote $\Delta\bar{L}_i = \bar{L}_i^{-} - \bar{L}_i^{+}$ and $\Delta L_i^i = L_j^{i-} - L_j^{i+}$. Using a similar development as in the proof of Theorem 3.1, we obtain the following refined bound.

**Theorem G.1.** *Assume $\ell(\cdot,\cdot) \in [0,1]$, then we have*

$$|\mathcal{E}_{\mathcal{D}}(\mathcal{A})| \leq \frac{1}{K}\sum_{i=1}^{K}\sqrt{2I(\Delta\bar{L}_i;V_i)} + \frac{1}{Kn}\sum_{i=1}^{K}\sum_{j=1}^{n}\mathbb{E}_{V_i}\sqrt{2I^{V_i}(\Delta L_j^i;U_j^{i,V_i})}.$$

*Proof Sketch.* Following the proof of Theorem 3.1, the function $g(\tilde{z}^i,u^i,w,v_i) = (-1)^{v_i}\left(\ell(w,\tilde{z}_{1,u_1^{i,1}}^{i,1}) - \ell(w,\tilde{z}_{1,u_1^{i,0}}^{i,0})\right)$ transforms into

$$g(\Delta\bar{L}_i,V_i) = (-1)^{V_i}\Delta\bar{L}_i,$$

Similarly, the function $h(\tilde{z}^i,v_i,w,u_j^{i,v_i}) = (-1)^{u_j^{i,v_i}}\left(\ell(w,\tilde{z}_{j,1}^{i,v_i}) - \ell(w,\tilde{z}_{j,0}^{i,v_i})\right)$ now becomes

$$h(V_i,\Delta L_j^i,U_j^{i,V_i}) = (-1)^{U_j^{i,V_i}}\Delta L_j^i.$$

The remaining steps follow the same structure as the original proof, with the expectation now taken over the newly defined loss difference random variables. $\square$

By data-processing inequality and chain rule, Theorem G.1 is tighter than Theorem 3.1.

## H. Additional Experiment Details

We adapt the code from https://github.com/hrayrhar/f-CMI for supersample construction and CMI computation, and we use the FL training code from https://github.com/vaseline555/Federated-Learning-in-PyTorch. We now state the complete experiment details below.

**MNIST**  We train a CNN with approximately 170K parameters. The model consists of two $5 \times 5$ convolutional layers—the first with 32 channels and the second with 64—each followed by $2 \times 2$ max pooling. These are followed by a fully connected layer with 512 units and ReLU activation, and a final softmax output layer. Each local training algorithm $\mathcal{A}_i$ trains this model using full-batch GD with an initial learning rate of 0.1, which decays by a factor of 0.01 every 10 steps. At each FL round, clients train locally for 5 epochs before sending their models to the central server. The entire training process spans communication 300 rounds between clients and the central server, reducing the commonly used 1000 rounds to lower computational costs.

**CIFAR-10**  We use the same CNN model from McMahan et al. (2017), which has approximately $10^6$ parameters. The architecture consists of two convolutional layers, followed by two fully connected layers and a final linear transformation layer. The CIFAR-10 images are preprocessed as part of the training pipeline, including cropping to $24 \times 24$, random horizontal flipping, and adjustments to contrast, brightness, and whitening. Each local training algorithm $\mathcal{A}_i$ trains the CNN model using SGD with a mini-batch size of 50 and follows the same learning rate schedule as in the MNIST experiment. As in the MNIST setup, clients train locally for five epochs per round before sending their models to the central server, with training spanning 300 communication rounds.

**Non-IID Setting and Evaluation Metric**  For both of the classification tasks, we evaluate prediction error as our performance metric, i.e. we utilize the zero-one loss function to compute generalization error. During training, we use cross-entropy loss as a surrogate to enable optimization with gradient-based methods. To introduce a non-IID setting, we apply a pathological non-IID data partitioning scheme as in McMahan et al. (2017). Specifically, all images are first sorted by their labels, divided into shards, and then assigned to clients such that each client receives two shards, resulting in most clients having examples from only two digit classes. Additionally, since we pre-define the non-participating clients from the superclient, all participating clients are included in every round of training.

**CMI Estimation**  For supersample and superclient construction, when analyzing generalization behavior concerning the sample size $n$, we fix the superclient size at 100, leading 50 participating clients and 50 non-participating clients randomly selected by $V$. The sample size per client varies within $n \in \{10, 50, 100, 250\}$. When analyzing generalization behavior with respect to the number of participating clients $K$, we set $n = 100$ for MNIST and $n = 50$ for CIFAR-10, varying the number of clients as $K \in \{10, 20, 30, 50\}$. To estimate the CMI terms, for both tasks, we draw three samples of $\tilde{Z}$ and $V$, and 15 samples of $U$ for each given $\tilde{z}$ and $v$. In each experiment, for a fixed sample size and client number, the individual CMI term $I(L_j^{i+}; U_j^{i,V_i}|V_i = v_i)$ is estimated using 15 samples, while $I(\bar{L}_i^+; V_i)$ is estimated using 45 samples, utilizing a simple mutual information estimator for discrete random variables as in Harutyunyan et al. (2021). In total, we conduct 720 runs of FedAvg for these classification tasks. All experiments are performed using NVIDIA A100 GPUs with 40 GB of memory.

