# OpenReview forum: "Generalization in Federated Learning: A Conditional Mutual Information Framework"
_ICML.cc/2025/Conference — ICML 2025 poster_

### Official Review · Reviewer_k9pd · 2025-03-10

**Overall Recommendation:** 4

**Summary:**

The paper introduces a novel information-theoretic framework to analyze generalization in federated learning (FL). By extending the supersample-based conditional mutual information (CMI) framework with a “superclient” construction, the authors decompose the generalization error into two components: the participation gap (between participating and non-participating clients) and the out‐of‐sample gap (local generalization error). They derive multiple CMI-based bounds, including high-probability and excess risk bounds, and show that differential privacy constraints naturally lead to tighter generalization guarantees. Experiments with FedAvg demonstrate that the evaluated CMI bounds are non-vacuous and reflect actual generalization behavior.

## Update after rebuttal
I will stay at my current score and recommend a (4): Accept

**Claims And Evidence:**

The paper makes the following claims:
- The superclient construction yields a two-level decomposition of generalization error with tight CMI-based bounds.
- Differential privacy at both local and global levels ensures that CMI terms (and thus the generalization error) remain bounded.
- Evaluated CMI (e-CMI) bounds can recover best-known FL convergence rates in low empirical risk regimes.

These claims are well supported by both detailed theoretical proofs (Theorems 4.1, 4.2, and 4.3) and empirical evaluations using standard FL frameworks such as FedAvg (discussed in Section 7).

**Essential References Not Discussed:**

While the paper references key works in CMI and FL, a discussion comparing the new bounds with very recent developments in PAC-Bayesian approaches or alternative stability-based analyses could further contextualize the contributions.

**Experimental Designs Or Analyses:**

The experimental work supports the theory; however, the paper could be improved by testing on more diverse datasets or different federated learning scenarios to provide a broader view of the approach’s effectiveness.

This suggestion is not a weakness (in terms of publication) but an opportunity for further validation.

**Methods And Evaluation Criteria:**

The paper extends the existing conditional mutual information framework to federated learning. It introduces a new "superclient" construction along with supersamples to address the two levels of generalization error. The methods are rigorous, employing tools like KL divergence and concentration inequalities.

While the mathematical methods are solid, including more straightforward explanations or diagrams could help clarify the concepts for readers who may not be experts in the field.

**Other Comments Or Suggestions:**

I do not have any major comments, and only have a few suggestions. Consider including a more detailed discussion of potential limitations or assumptions (e.g., reliance on bounded loss functions or the i.i.d. assumption in some cases). Another suggestion is to enhance the clarity of the presentation, perhaps through more diagrams or flowcharts, which would help readers unfamiliar with CMI techniques.

**Other Strengths And Weaknesses:**

**Strengths:**
- Innovative extension of the CMI framework with a superclient construction, addressing a gap in FL generalization analysis.
- Rigorous theoretical derivations that are well-connected to existing literature.
- Empirical validation that supports the theoretical claims.

**Weaknesses:**
- No major weaknesses were identified. The experimental section could be expanded to include more diverse FL settings or real-world datasets for broader validation

**Questions For Authors:**

- How sensitive are the bounds to deviations from the bounded loss assumption in practice?
- Could you provide further insight into how the e-CMI bounds might be estimated in high-dimensional settings beyond the one-dimensional case?
- How do the derived bounds perform in non-i.i.d. client scenarios that are common in cross-device FL?

Providing answers to these questions may help readers better understand and apply this framework to analyze their FL algorithms.

**Relation To Broader Scientific Literature:**

The work builds on established information-theoretic generalization analyses and extends them to the federated learning setting. It relates to recent advances in FL generalization and connects with the broader literature on differential privacy and meta-learning.

**Theoretical Claims:**

The authors make strong theoretical claims by breaking down the generalization error into two parts: the participation gap and the out-of-sample gap. They also show that differential privacy helps to keep these errors small. The detailed proofs build on well-known ideas from the literature.

It would be beneficial if the paper discussed any limitations of these theoretical results, such as their dependence on the bounded loss function assumption.

---

> ### Author Rebuttal · Authors · 2025-03-31
>
> We thank you sincerely for your valuable feedback on our paper. Our responses follow.
>
> >- While the mathematical methods are solid, including more straightforward explanations or diagrams could help clarify the concepts for readers who may not be experts in the field.
>
> >- Another suggestion is to enhance the clarity of the presentation, perhaps through more diagrams or flowcharts, which would help readers unfamiliar with CMI techniques.
>
> **Response.** Thank you for your suggestion. We have included a visualization of our superclient and supersample construction, and we will also provide additional background on CMI techniques along with more accessible explanations of these concepts in the revised version.
>
>
> >- It would be beneficial if the paper discussed any limitations of these theoretical results, such as their dependence on the bounded loss function assumption.
> >-  Consider including a more detailed discussion of potential limitations or assumptions (e.g., reliance on bounded loss functions or the i.i.d. assumption in some cases).
> >- How sensitive are the bounds to deviations from the bounded loss assumption in practice?
>
> **Response.** Regarding the boundedness assumption, as noted in the paragraph immediately following Theorem 6.2 (Lines 361–362) in the paper, this assumption can be relaxed to a sub-Gaussian condition, where the loss function may be unbounded but exhibits Gaussian-like tail behavior. Under this relaxed assumption, all the theoretical results in our paper remain valid. In the case of heavy-tailed losses, additional techniques such as truncation would be required to ensure the results hold. We will elaborate further on these points in the revised version.
>
> >- The experimental work supports the theory; however, the paper could be improved by testing on more diverse datasets or different federated learning scenarios to provide a broader view of the approach’s effectiveness.
>
> **Response.** Thank you very much for the suggestion. We will include an additional dataset in the revised version, and if the reviewer has any specific dataset in mind, we would be happy to incorporate it.
>
> >- While the paper references key works in CMI and FL, a discussion comparing the new bounds with very recent developments in PAC-Bayesian approaches or alternative stability-based analyses could further contextualize the contributions.
>
> **Response.** Compared to PAC-Bayesian and stability-based bounds, our e-CMI bounds are significantly easier to estimate in practice. Moreover, existing PAC-Bayesian and stability-based bounds do not account for the participation gap and do not exhibit fast-rate behavior for overall generalization. We will include a more detailed discussion comparing our results with these existing bounds in the revised version.
>
> >- Could you provide further insight into how the e-CMI bounds might be estimated in high-dimensional settings beyond the one-dimensional case?
>
> **Response.** We note that the e-CMI bounds in our paper always involve mutual information between one-dimensional random variables by construction. Specifically, e-CMI refers to the mutual information between a single loss value (treated as a one-dimensional random variable, since the loss function maps inputs to real values) and a Bernoulli mask, which is a binary random variable. This low-dimensional structure makes e-CMI particularly easy to estimate, which is one of its key advantages.
>
>
> >- How do the derived bounds perform in non-i.i.d. client scenarios that are common in cross-device FL?
>
> **Response.** Our bounds hold under a non-i.i.d. setting in the sense that each client may have a different data distribution, which is typical in cross-device FL. However, we do assume that clients are sampled independently, and that within each client, data points are drawn i.i.d. If either of these assumptions does not hold, the generalization bounds would need to be refined using additional techniques to account for the dependencies, for example, by invoking graph-based dependencies among clients or modeling temporal (mixing) dependencies in the data. We will include this discussion in the revised version.

---

### Official Review · Reviewer_4cnC · 2025-03-10

**Overall Recommendation:** 2

**Summary:**

The paper studies the generalization error of the federated learning algorithm using the CMI framework. The goal is to capture the out-of-sample gap and the participation gap using this framework.  To do so, a federated learning setup is considered where each user observes data distributed according to a distribution sampled from a meta-distribution $\mathcal{D}$.

The established bounds, including the fast-rate bounds, include two terms that capture the two above-mentioned gaps: the first term captures the effect of uncertainty in the choice of distribution, while the second term is a conditional mutual information term, as derived in the paper of Zakynthinou and Steinke.

**Claims And Evidence:**

1. The paper claims to study the generalization error of FL. However, I believe that the bounds established do not capture any characteristic of FL algorithms. In particular,

   - The results obtained in Sections 4 and 5 are valid for "any" learning algorithm, and not necessarily for a federated learning algorithm, since in the end what is considered is the overall learning algorithm $\mathcal{A}:\mathcal{Z}^{nK} \to \mathcal{W}$, as a black box. Therefore, in my opinion, such results do not reflect any aspect specific to FL.

   - The results of section 5 are valid for one-round aggregation and under simplifying assumptions (such as Bregman divergence). Therefore, they cannot be considered as FL.

2. The claims about the behavior of Theorem 6.2. with respect to data heterogeneity do not seem to be precise enough. See questions.

**Essential References Not Discussed:**

The paper sufficiently addresses and discusses similar work. However, the precise advantage of the bounds obtained over the previous literature is not clear. More precisely, it is not clear what is the main question studied in this work.

**Experimental Designs Or Analyses:**

The experimental results are limited, but I do not think this is the main weakness of the paper.

**Methods And Evaluation Criteria:**

While CMI framework seems interesting; but I could not get the main take-home message of the paper. See the questions.

**Other Comments Or Suggestions:**

I found the notation very heavy and hard to follow. However, this may be partly unavoidable due to the complicated setup considered in this work.

**Other Strengths And Weaknesses:**

Strengths:

1. The issue of studying the generalization error of FL is important, and the use of the CMI framework looks interesting.
2. A number of previously established results are adapted to this setup.
3. The idea of considering supersamples, where each client's "test" dataset may have a different distribution (sampled independently from the meta-distribution $\mathcal{D}$) than that client's training dataset distribution, to capture the effect of non-participating clients is interesting.

Weaknesses:
1. While the established bounds are correct, their proof techniques are rather standard extensions of CMI results to the distributed setup (similar to what has been considered in a number of papers for the mutual information based bounds) or to the setup where the distribution of each client comes from a metadistribution. The latter case is also not new; as considered for example in Theorem 5.1 in Zhang et al. (Neurips 2024). The established results are mainly an adaptation of the previous results; without any significant novelty.
2. Many order-wise behaviors are not rigorous. See questions.

**Questions For Authors:**

1.	My main concern is with the implications of the bounds. What are the new conclusions we learn from these bounds for FL? What exactly is the message of this paper? I think the general argument that MI bounds can become vacuous but CMI bounds can never become unbounded is not sufficient for a new publication. The authors need to give concrete examples or study cases where their result brings a new insight or understanding.

2.	In various places it is mentioned that the order is $\mathcal{O}\left(\frac{1}{\sqrt{K}}+\frac{1}{\sqrt{Kn}}\right)$. However, if the order-wise behavior is examined, then $\mathcal{O}\left(\frac{1}{\sqrt{K}}+\frac{1}{\sqrt{Kn}}\right)= \mathcal{O}\left(\frac{1}{\sqrt{K}}\right)$. So the order-wise behavior of the bounds does not depend on $n$, which is a bad sign. Can you explain this?


3.	Following on from the above point, it seems that in various places the above conclusions implicitly assume that KL divergences (or mutual information) are $\mathcal{O}\left(1\right)$, which is certainly not true in general. Therefore, the above order-wise behavior is not correct.

4.	It is claimed that the bound of Theorem 6.2 becomes larger for more heterogeneous data. More precisely, it is mentioned that in the non-interpolating version of this result, the bound is proportional to $\mathbb{E}[\ell(W,Z_{i,j})]-\mathbb{E}[\ell(W_i,Z_{i,j})]$, and since this term becomes larger for more heterogeneous data, the bound then increases with heterogeneity. However, this argument is not sufficient because the second term of the bound is the square root of the product of this term and the conditional mutual information term. To evaluate the behavior of the bound with respect to heterogeneity, the behavior of this CMI term as a function of data heterogeneity must also be studied.

**Relation To Broader Scientific Literature:**

Sufficiently discussed.

**Theoretical Claims:**

The established results sound correct to me.

---

> ### Author Rebuttal · Authors · 2025-03-31
>
> We thank you sincerely for your valuable feedback on our paper. Our responses follow.
>
> >-... the bounds established do not capture any characteristic of FL ...
>
> **Response.** We respectfully disagree with the reviewer's claim that our bounds fail to capture key characteristics of FL, and the argument that the overall algorithm is ultimately treated as a black box is a misinterpretation of our results in Sections 4 & 5. To analyze FL in the two-level generalization setting using CMI, the introduction of the superclient is essential—a construction not needed in centralized learning and one that highlights the fundamental differences between the two settings. Moreover, treating the FL algorithm as a global mapping and construct a supersample of size $2nK$, would break the symmetry required for CMI analysis due to the non-i.i.d. nature.
>
> Instead, we believe the more accurate interpretation is that our results in Sections 4 & 5 are designed to apply to any FL algorithm, without being restricted to a particular instantiation. The goal of these sections is to establish a general framework for analyzing FL as a learning problem. This generality should not be viewed as a weakness.
>
> In Section 6, we go beyond this generality by considering specific aggregation strategies and loss functions, allowing us to derive sharper insights such as faster learning rates and the benefits of interpolation in local training. Although we focus on the one-round setting, as noted in Remark 6.1, extending the results to the multi-round case is straightforward. We plan to include the multi-round  extension in the Appendix, as introducing it in the main text would further complicate the already dense notation.
>
> >- Main take-home message ... | main question ...|Q1. My main concern ...
>
> **Response.** The main goal of our paper is to frame FL as a learning problem and analyze its learning rate in a two-level setting using CMI. Without assuming any specific FL strategy, our general two-step CMI bounds obtain several insights: (1) strong client privacy (both globally and locally) implies generalization; (2) in homogeneous settings, the bounds reduce to standard CMI analysis, with $I(W; V | \widetilde{Z}, U)$ capturing data heterogeneity; and (3) we identify conditions for fast learning rates.
>
> When applied to specific algorithms like FedAvg under certain loss assumptions, we show that FL can achieve even faster rates, thanks to model averaging and favorable loss properties. We believe this behavior extends to broader loss classes, though a formal proof remains open.
>
> Compared to existing bounds for FL, our e-CMI bound is notably easier to estimate and is the first  MI-based bound to give fast-rate guarantees for both the participation and out-of-sample gaps.
>
> Lastly, we refer the reviewer to our first response to Reviewer 4mA5 for a discussion on the practical implications of our results, and the difference between using generalization bounds as indicators of learnability v.s. as sufficient conditions for generalization performance.
>
> >- their proof techniques are rather standard ...
>
> **Response.** We acknowledge that, following our superclient construction, our CMI bounds build on existing techniques, with some extensions to accommodate the two-level setting. However, developing a general CMI framework for FL does not require a complete overhaul of prior methods. As noted in the review guidelines, "originality may arise from creative combinations of existing ideas", and we believe this applies to our contribution.
>
> >- Q2. ... order-wise behavior is not correct.
>
> **Response.**We believe the reviewer raises a valid point regarding order-wise behavior, and we also share the concern that many prior works overlook MI, KL, or other complexity terms in such analyses. In the absence of clear decay rates, we think the stated rates should be seen as reflecting worst-case behavior. For example, $O(1/\sqrt{K} + 1/\sqrt{Kn}) = O(1/\sqrt{K})$ implies that in highly heterogeneous settings, even with infinite local data ($n \to \infty$) for participating clients, FL may still fail to generalize to unseen clients—an intuitive outcome given the impact of extreme heterogeneity.
>
> Regardless of the exact decay of the CMI term, removing the square-root dependency remains desirable for faster convergence. We will clarify this in the revision.
>
> >- Q4. It is claimed that the bound ...
>
> **Response.** This seems to be a misinterpretation of our result. The CMI term in Theorem 6.2, unlike those in Sections 4 & 5, is a local CMI for each client, involving the local model $W_i$ rather than the global model $W$, and thus is not influenced by client heterogeneity.
>
> However, the reviewer’s intuition holds in a multi-round extension of Theorem 6.2, where $W_i$ may depend on other clients through repeated communication. We will clarify this in the revised version.

---

> > ### Comment · Reviewer_4cnC · 2025-04-01
> >
> > Thank you for the provided response. I believe the paper has some interesting results, but not enough.
> >
> > About capturing the characteristics of FL: It is not a misinterpretation. The bound explicitly depends only on the data distribution (which consists of K sets, each containing n samples i.i.d. from a distribution that is itself randomly chosen from a meta-distribution); not explicitly on the learning algorithm. Surely, it depends implicitly through the CMI terms, but this cannot be considered as capturing the FL characteristics since, otherwise, one could argue that [SZ20]'s bound also captures the effect of FL (when clients' data have the same distributions). The learning algorithm $\mathcal{A}:\mathcal{Z}^{nK} \to \mathcal{W}$ is indeed a black box; if the $nK$ data points are processed at one point or distributed, the bound holds. The authors probably see this as a strength, but I see it as a weakness: if a bound holds for both centralized and distributed learning algorithms, it cannot capture the characteristics of FL in my opinion.
> >
> > Hence, in my opinion, the bound cannot have an FL-specific take-home message. As mentioned above, the meta-learning point was already shown in Theorem 5.1 of Zhang et al. (Neurips 2024). Here is the CMI version of it (we know technically their proofs are not very different); but I am not convinced of any significant new result/message/conclusion in this paper.
> >
> > Regarding the orderwise analysis, I agree with the authors that such a loose orderwise discussion has unfortunately become common in many papers and is not specific to this paper.
> >
> > Finally, regarding Q4, again this is not a misinterpretation. To explain it better, consider the experiment performed by Sun et al. 2024. To understand the effect of heterogeneity, they introduced a family of distributions over the MNIST dataset, indexed by $\rho \in [0,1]$, where $\rho = 0$ corresponds to the homogeneous case and $\rho = 1$ is the most heterogeneous case. Importantly, for each $\rho$, the distribution of **all** clients changes. There is a rational behind this choice: to compare the performance of two sets of distributions (for clients), they should be comparable in some sense. In that paper, the setup is considered such that for each $\rho$, the marginal distribution over all clients for each digit remains as $1/10$. Thus, we can now see that, for example, to apply Theorem 6.2 to this setup, the local CMI terms also change since the distribution of each client's data changes. Let's, for simplicity, first assume that instead of local CMI terms in Theorem 6.2, we had MI terms, i.e. $I(S;W_i)$ (or their signle-datum version), since their analysis is simpler in this case. Then, in the case where a similar algorithm is used in all clients, due to the concavity of $I(S;W_i)$ with respect to the data distribution, $\frac{1}{K}\sum_i I(S;W_i)$ for the heterogeneous case is smaller than $\frac{1}{K}\sum_i I(S;W_i)$ for the homogeneous case. I think a similar result holds for CMI terms as well (I am not sure, but if not, it needs to be shown). Thus, in a suitable setup, they are not easily comparable using Theorem 6.2., even if the CMI term includes only local algorithms.

---

> > > ### Author Response · Authors · 2025-04-07
> > >
> > > We sincerely thank the reviewer for their prompt reply and for actively engaging in the discussion.
> > >
> > > >- The bound explicitly depends only on the data distribution ...  if a bound holds for both centralized and distributed learning algorithms, it cannot capture the characteristics of FL in my opinion.
> > >
> > > We thank the reviewer for further clarifying the concern. However, we still feel that the argument *"if a bound holds for both centralized and distributed learning algorithms, it cannot capture the characteristics of FL"* is a rather subjective point of view. In particular, if centralized learning can be viewed as a special case of distributed learning (where all data resides on a single client), then a generalization bound that holds for both centralized and distributed settings should be considered a desirable property, not a limitation.
> > >
> > > For example, in the work of Sun et al. (2024), the last sentence of the abstract states: "Particularly, in the i.i.d. setting, our results recover the classical results of stochastic gradient descent (SGD)". Moreover, the third contribution listed in their Section 1.2 emphasizes that "In i.i.d. setting with convex loss functions, our bounds match existing results of SGD in the sense that FedAvg reduces to SGD method". This clearly demonstrates that their bounds hold for both centralized (SGD) and distributed (FedAvg) cases. Yet, it may not be reasonable to claim that their bound fails to capture the characteristics of FL.
> > >
> > > Similarly, since the standard stability-based generalization bound for SGD is considered as a special case of the results in Sun et al. (2024), it is completely acceptable to say that [SZ20]'s bound captures the effect of FL in the i.i.d. case (i.e. Corollary 4.2), and our results generalize [SZ20].
> > >
> > > We would also like to clarify that to rigorously study the characteristics of any specific FL algorithm, establishing a general generalization framework for FL is a necessary step. This is precisely how we organize our paper: Sections 4 and 5 present a general framework that treats FL as a learning problem without committing to any specific algorithm.  This level of generality is what enables us to analyze specific FL settings in Section 6. In other words, if the generality of Sections 4 and 5 is viewed as a weakness, it would be difficult for us to further improve the paper, as this general foundation is essential to our overall contribution.
> > >
> > > >- ... the meta-learning point was already shown in Theorem 5.1 of Zhang et al. ... but I am not convinced of any significant new result/message/conclusion in this paper.
> > >
> > > As noted, our paper treats FL as a learning problem and mainly focus on its learning rates and learnability. Accordingly, we study the tightness of the generalization bound (e.g., fast-rate bounds), high-probability guarantees, and the conditions under which FL remains learnable (e.g., privacy) and achieves sharper learning rates (e.g., model averaging under strictly convex and smooth losses). In contrast, Zhang et al. (2024) take a practitioner-centric perspective, seeking a generalization guarantee for their specific algorithm and thus invoking a looser input-output MI bound. However, using the existence of their work to cast a negative light on our broader learning-theoretic contributions overlooks the full scope of our results.
> > >
> > > >- Finally, regarding Q4, ...
> > >
> > > We appreciate the reviewer's efforts in further clarifying the question. Regarding the relationship between the CMI terms in Theorem 6.2 and data heterogeneity, we suggest focusing on the fundamental quantity that our CMI term is used to bound, namely, the local generalization gap for client $i$, expressed as $\mathbb{E}[\ell(W\_i,Z'\_{i,j})-\ell(W\_i,Z\_{i,j})]$ (see Eq. (48) or Eq. (47) in Appendix). In our single-round setting, the local model $W\_i$ is trained independently and has not communicated with other clients, so this local generalization gap is, by construction, independent of any other clients.
> > >
> > > Initially, it seems counterintuitive to us to describe this quantity as a function of data heterogeneity, since it is only about a single client's data and model. In this context, analyzing the local generalization gap alone seems unrelated to the notion of data heterogeneity across clients. Based on the reviewer's further explanation, we now understand that if the data distribution itself is governed by some data heterogeneity parameter, then the value of this local gap may indeed vary with that parameter since the gap is ultimately a function of the underlying data distribution.
> > >
> > > That said, we note that even in homogeneous (i.i.d.) settings, different data distributions can result in different values of the local generalization gap. Therefore, while changes in the CMI terms may be influenced by variations in data heterogeneity, they do not, in general, give a direct or unique measure of it. We acknowledge this subtlety and will clarify it in the revised version of the paper.

---

### Official Review · Reviewer_4mA5 · 2025-03-15

**Overall Recommendation:** 3

**Summary:**

The paper proves mutual information-based generalization bounds for a federated learning setting, that bounds both the out-of-sample gap (between the empirical and population distributions of the participating clients) and the participation gap (between the participating clients and the underlying meta distribution of clients). Theorem 4.1 includes the generalization bound as the sum of the bounds for the terms of out-of-sample and participation gaps, which is order-wise bounded in Remark 4.2 assuming a constant bound on the differential privacy degree of the FL algorithm. Next, in Theorem 4.2 the authors present a PAC-style generalization bound that bounds the gap with high probability. In section 5, the authors show fast rate bounds that decay as $ \mathcal{O} \left( \frac{1}{\sqrt{nK}}+\frac{1}{\sqrt{K}} \right) $ for $K$ clients and $n$ samples per client. Section 7 discusses some numerical results for FedAvg trained models, which show the smaller generalization gap when increasing $K$ or $n$.

**Claims And Evidence:**

The paper's theoretical claims on generalization bounds for federated learning seem correct. The current main text contains many theorems, which leads to less space for discussing the implications of the theorems. It would be better to dedicate more space for discussing why the MI-based generalization analysis for federated learning will be useful and how the approach can lead to regularization methods for reducing the out-of-sample gap and participation gaps.

In addition, I would like to ask the authors how the theoretical analysis in this paper goes beyond a direct application of the existing MI generalization bounds to the FL setting. I understand that the FL problem has two error terms to bound (out-of-sample and participation gaps). Still, the authors seem to apply the MI generalization bounds in a standard centralized-like seting to bound each of the error terms. Can the authors elaborate on how their analysis contributes beyond the existing MI-based generalization frameworks?

**Essential References Not Discussed:**

Yes

**Experimental Designs Or Analyses:**

The experiments in Section 7 provide a reasonable sanity check that the bounds correlate with actual generalization error. However, the authors do not separately report the "out-of-sample gap" and "participation gap." I wonder how fast the participation gap (when considered alone) decreases with $K$. Also, the non-IID setting should be explained in more detail (in the supplementary lines 1666-1672 several details on the frequencies and parameters are missing), and the separated out-of-sample and participation gaps should be reported to see how they change with $K$ and $n$.

**Methods And Evaluation Criteria:**

There is little discussion on estimating the mutual information terms in the generalization upper bounds when evaluating the generalization bounds in section 7. I want to ask the authors whether the bounded sample size of the clients is enough to estimate the mutual information terms in the generalization upper-bound. It seems to me that a challenge with MI generalization bounds is how to have a tight estimation of the mutual information term in the bound for the high-dimensional variables in the neural net layers.

**Other Comments Or Suggestions:**

See the previous comments.

**Other Strengths And Weaknesses:**

See the previous comments.

**Questions For Authors:**

See the previous comments.

**Relation To Broader Scientific Literature:**

While the paper extends mutual information-based generalization analysis to federated learning, I am uncertain about the practical role of the proposed generalization bounds in improving FL algorithms. In statistical learning theory, generalization bounds often introduce a capacity norm or an implicit property of the hypothesis class, which can be explicitly or implicitly regularized to reduce the generalization gap. However, in this work, it is unclear whether the MI-based generalization bounds can translate into regularization methods to improve generalization.

Can the authors clarify how their generalization bounds can be used to regularize and reduce the participation gap in FL? The numerical results seem to suggest that the only way to reduce the gap is to increase the number of samples and clients.

**Theoretical Claims:**

Not in detail, but the results seem correct to me. As I asked before, can the authors explain what extra challenges their analysis addresses beyond the existing MI generalization results that can be applied to bound each of the out-of-sample and participation gaps?

---

> ### Author Rebuttal · Authors · 2025-04-01
>
> We thank you sincerely for your valuable feedback on our paper. Our responses follow.
>
> >- ... why the MI-based ... will be useful and how ... lead to regularization ...
> >- ... uncertain about the practical role ...
>
> **Response.** We note that our bounds do have practical implications, e.g., as mentioned in the Related Work section (Line 84-90), if the CMI terms in Theorem 6.2 are small, then the norm $||w-w_i||$  is also small, which is the regularization term used in FedProx. Upon reflection, we realize that this discussion would be more appropriately placed directly after Theorem 6.2, to make the implication more explicit.
>
> We understand that the reviewer may have been looking for new regularization schemes for reducing participation gap and out-of-sample gap, rather than connections to well-known ideas such as norm-based capacity control. In response, we recommend focusing on Theorem C.2 (which, while looser than the CMI bounds, is more interpretable). This result will ultimately lead to a gradient-norm-based regularizer when SGD or SGLD is used, and hints at a potential feature alignment mechanism in the KL sense for clients. We will elaborate on these implications in the revised version.
>
> Finally, regarding the practical implication, we would like to share a perspective based on our own research experience in learning theory. If the goal is to derive sharper generalization bounds with fast decay rates, i.e. to study learnability and learning rates of a problem, then practical applications may not follow directly. In this case, generalization bounds mainly serve to evaluate whether learning is theoretically possible. In the extreme case, the tightest generalization bound is the generalization error itself, which provides no new actionable insights for algorithm design. If the goal is to obtain actionable insights for algorithm design, then tightness of the bound becomes less critical. Generalization upper bounds, even if loose, can serve as a basis for designing regularizers. For example, penalizing weight norms is a widely used practice to improve generalization, despite the well-known fact that norm-based bounds are often vacuous and cannot explain generalization behavior in deep learning [R1, R2].
>
> [R1] Vaishnavh Nagarajan, and J. Zico Kolter. "Uniform convergence may be unable to explain generalization in deep learning." NeurIPS 2019.
>
> [R2] Yiding Jiang, et al. "Fantastic Generalization Measures and Where to Find Them." ICLR 2020.
>
> >- ... how the theoretical analysis ...?
> >- ... explain what extra challenges ...?
>
> **Response.** The bounding steps for the out-of-sample gap are indeed similar to those in standard centralized analysis, which is expected as each individual out-of-sample gap is close to an in-distribution generalization gap. The main challenges arise in bounding the participation gap. Compared to existing MI-based bounds, our contributions include: the construction of the superclient, the proof of its symmetry properties (Lemma 4.1), the shifted Rademacher representation of the weighted participation gap (Lemma E.1), and the leave-one-out argument for the participation gap (Lemma F.1). To clarify, the use of weighted generalization error and leave-one-out arguments is not new in the broader learning theory literature. However, their application in our setting is enabled by the superclient construction, which serves as the foundation for these results.
>
> >- ... on estimating the mutual information ...
>
> **Response.** Please notice that we use e-CMI bounds in experiments to avoid the difficulties of estimating MI between high-dimensional R.V.'s. The second sentence in Section 7 states: "we estimate the fast-rate e-CMI bound for FL, as… Additionally, due to the challenges associated with estimating MI when dealing with high-dimensional random variables, we compute an evaluated version of the CMI bound from Theorem 4.1".  Importantly, the e-CMI bound is computed between two one-dimensional variables (one of which is binary), making the estimation easy and eliminating the need for advanced MI estimators. Further details are in Appendix H.
>
> >- ... do not separately report ...
> >- ... the non-IID setting ... and the separated ...
>
> **Response.** The pathological non-IID data partitioning follows McMahan et al. (2017): data are sorted by label, split into 200 shards of size 300, and each client is randomly assigned 2 shards. We will include more details in the revision.
>
> As for separate plots of the participation gap (PG) and out-of-sample gap (OG), we would like to clarify that our experiments are based on the e-CMI bound in Theorem 5.1, which consists of three components weighted by jointly optimizable coefficients $C_1, C_2, C_3, C_4$. These coefficients are optimized using the SLSQP algorithm implemented in the SciPy package. As a result, the e-CMI bound is not a simple addition of the individual bounds for PG and OG, which makes it difficult to present separate plots for these two quantities.

---

> > ### Comment · Reviewer_4mA5 · 2025-04-05
> >
> > I thank the authors for their detailed response to my comments and questions. I find the responses satisfactory and now can better appreciate the authors' motivation behind the CMI generalization bounds. I will update my score accordingly.

---

> > > ### Author Response · Authors · 2025-04-07
> > >
> > > We sincerely thank the reviewer for taking the time to carefully read our responses, and we are glad that our clarifications helped convey the motivation behind the CMI-based generalization bounds. We will incorporate the discussions from the rebuttal into the revised version of the paper.

---

### Official Review · Reviewer_XAJx · 2025-03-17

**Overall Recommendation:** 3

**Summary:**

This work studies the question of generalisation in federated learning (FL), where $K$ users aims to share some benefits of their learning phase via a central server without sharing their data. Authors propose novel generalisation bounds tailored to FL involving the Conditional Mutual Information (CMI) framework, yielding original information-theoretic results. Authors first focus on general in-expectation and high-probability generalisation bounds (Section 4), before proposing fast rates (Section 5). Finally, they involve the specificities of some popular aggregation algorithms used in FL in Section 6 and empirically study the tightness of their bounds in Section 7.

**Claims And Evidence:**

All theoretical results look reasonable and extend CMI bounds beyond batch learning to reach FL. The impact of their fast rate result compared to those of Section 4 is well established in Section 7.

Something which would gain to be clearer: you said in l.122-123 left column that your CMI framework is inspired by the meta-learning one of Hellstrom&Durisi 2022. Is it possible to precise what are the specificities of your derived results (e.g. those of Section 4) compared to theirs? It seems you control the same global true risk, due to the assumptions that all tasks are drawn according to $\mathcal{D}$.

**Essential References Not Discussed:**

I do not know enough about either the FL or CMI literature to provide relevant feedback.

**Experimental Designs Or Analyses:**

I did not check in detail the experimental protocol.

**Methods And Evaluation Criteria:**

The experimental framework looks sound, with reasonably big CNNs (170K parameters), which is nice for a theoretical paper althouigh I did not check carefully the details.

An important point: it seems that there is no comparison with existing bounds. For instance, how does your bound behave wrt those of, e.g., Sefidgaran et al. 2024? Is it challenging to plot their results ?

**Other Comments Or Suggestions:**

None.

**Other Strengths And Weaknesses:**

None.

**Questions For Authors:**

- Something which would gain to be clearer: you said in l.122-123 left column that your CMI framework is inspired by the meta-learning one of Hellstrom&Durisi 2022. Is it possible to precise what are the specificities of your derived results (e.g. those of Section 4) compared to theirs? It seems you control the same global true risk, due to the assumptions that all tasks are drawn according to $\mathcal{D}$.
- It seems that there is no comparison with existing bounds. For instance, how does your bound behave wrt those of, e.g., Sefidgaran et al. 2024? Is it challenging to plot their results ?
- In most of your results, you have a $O(1/\sqrt{K})$ term. Would it be possbile to recover the influence of $n$ in such terms (maybe through the mutual information term)?


In conclusion, this work looks serious, theoretically well-grounded with plethora of new generalisation bounds and nice experiments (which would gain to be completed). However, I do not know much about either FL or CMI literature.

**Relation To Broader Scientific Literature:**

I do not know enough about either the FL or CMI literature to provide relevant feedback.

**Theoretical Claims:**

I only looked at the proof of Theorem 4.1, which seems correct.

---

> ### Author Rebuttal · Authors · 2025-03-31
>
> We thank you sincerely for your valuable feedback on our paper. Our responses follow.
>
> >- you said in l.122-123 left column that your CMI framework is inspired by the meta-learning one of Hellstrom \&Durisi 2022. Is it possible to precise what are the specificities of your derived results (e.g. those of Section 4) compared to theirs? It seems you control the same global true risk, due to the assumptions that all tasks are drawn according to $\mathcal{D}$.
>
> **Response.** As mentioned in the paper, our construction is indeed inspired by the framework for meta-learning proposed by Hellström & Durisi (2022). However, our results are not directly comparable to theirs due to a key difference in problem setup: their meta-learning framework requires the meta-learner (i.e., the global model $W$) to be further trained on the test tasks (i.e., previously non-participating clients), whereas in FL, the global model is evaluated directly on unseen clients without any additional local fine-tuning. To enable a direct comparison with Hellström & Durisi (2022), the CMI framework presented in this paper would need to be extended to the personalized FL setting, where the global model is allowed further local adaptation. In that case, Lemma 4.1 would also need to be revised accordingly, as the hypothesis may no longer be invariant to the ordering of the "test data". We will include these discussions in the next revision.
>
> >- An important point: it seems that there is no comparison with existing bounds. For instance, how does your bound behave wrt those of, e.g., Sefidgaran et al. 2024? Is it challenging to plot their results?
>
> **Response.** The PAC-Bayesian and rate-distortion bounds in Sefidgaran et al. (2024) are indeed challenging to compute numerically for more complex neural networks, as they require estimating KL or MI between high-dimensional random variables, even when the model parameters are quantized. Notably, in their paper, the generalization bounds are not plotted for their ResNet experiments on CIFAR-10; instead, they only present the behavior of the generalization error to support the insights behind their bounds. In contrast, a key advantage of our results lies in the ease of estimating the e-CMI bound in the FL setting, making it more practical for empirical evaluation.
>
> >- In most of your results, you have a $O(1/\sqrt{K})$ term. Would it be possbile to recover the influence of $n$ in such terms (maybe through the mutual information term)?
>
> **Response.** Yes, the reviewer raises a valid point. Indeed, the participation gap term seems to follow an $O(1/\sqrt{K})$ behavior. At the same time, in the i.i.d. setting, increasing $n$ is also expected to reduce the participation gap. This effect is implicitly captured by the CMI term $I(W;V|\widetilde{Z},U)$, as demonstrated in Corollary 4.2, where all clients share the same data distribution (i.e., there is only one client distribution). In the non-i.i.d. case, however, it is more challenging to explicitly characterize the impact of $n$ on the participation gap, as its quantitative effect depends on the degree of data heterogeneity, which can vary significantly across scenarios. We will include this discussion in the revised version.

---

### Decision · Program_Chairs · 2025-05-01

**Decision:**

Accept (poster)

**Comment:**

This paper introduces a novel theoretical framework based on Conditional Mutual Information (CMI) to analyze generalization in Federated Learning (FL), specifically addressing the two-level nature (out-of-sample and participation gaps) via a "superclient" construction. The work derives several CMI bounds within this framework and provides empirical estimations suggesting these bounds correlate with generalization behavior. The reviews present a split perspective: two reviewers strongly support the paper based on the theoretical novelty and framework extension, finding it a solid contribution despite potential looseness or lack of algorithm specificity. Conversely, two other reviewers express significant reservations, questioning the practical tightness, interpretability, and advantages over existing bounding techniques (like PAC-Bayes), while also criticizing the limited experimental scope and presentation clarity. The authors' rebuttal primarily defended the theoretical contribution and empirical methodology. While the concerns about the bounds' practicality and tightness are valid and characteristic of information-theoretic approaches, the consensus acknowledges the novelty of extending the CMI framework to FL's unique structure. Balancing the clear theoretical contribution against valid concerns about practical utility and experimental breadth, the paper appears to be a borderline case, leaning towards acceptance based on its novel framework for understanding FL generalization.